# Model-guided metabolic rewiring to bypass pyruvate oxidation for pyruvate derivative synthesis by minimizing carbon loss

Yun Zhang,[1] Xueliang Wang,[1,2] Christianah Odesanmi,[1,2] Qitiao Hu,[1,2] Dandan Li,[1] Yuan Tang,[1,2] Zhe Liu,[1,2] Jie Mi,[1,2] Shuwen Liu,[1] Tingyi Wen[1,3]

**ABSTRACT** Engineering microbial hosts to synthesize pyruvate derivatives depends on blocking pyruvate oxidation, thereby causing severe growth defects in aerobic glucose-based bioprocesses. To decouple pyruvate metabolism from cell growth to improve pyruvate availability, a genome-scale metabolic model combined with constraint-based flux balance analysis, geometric flux balance analysis, and flux variable analysis was used to identify genetic targets for strain design. Using translation elements from a ~3,000 cistronic library to modulate *fxpK* expression in a bicistronic cassette, a bifido shunt pathway was introduced to generate three molecules of non-pyruvate-derived acetyl-CoA from one molecule of glucose, bypassing pyruvate oxidation and carbon dioxide generation. The dynamic control of flux distribution by T7 RNAP-mediated synthetic small RNA decoupled pyruvate catabolism from cell growth. Adaptive laboratory evolution and multi-omics analysis revealed that a mutated isocitrate dehydrogenase functioned as a metabolic switch to activate the glyoxylate shunt as the only C4 anaplerotic pathway to generate malate from two molecules of acetyl-CoA input and bypass two decarboxylation reactions in the tricarboxylic acid cycle. A chassis strain for pyruvate derivative synthesis was constructed to reduce carbon loss by using the glyoxylate shunt as the only C4 anaplerotic pathway and the bifido shunt as a non-pyruvate-derived acetyl-CoA synthetic pathway and produced 22.46, 27.62, and 6.28 g/L of L-leucine, L-alanine, and L-valine by a controlled small RNA switch, respectively. Our study establishes a novel metabolic pattern of glucose-grown bacteria to minimize carbon loss under aerobic conditions and provides valuable insights into cell design for manufacturing pyruvate-derived products.

**IMPORTANCE** Bio-manufacturing from biomass-derived carbon sources using microbes as a cell factory provides an eco-friendly alternative to petrochemical-based processes. Pyruvate serves as a crucial building block for the biosynthesis of industrial chemicals; however, it is different to improve pyruvate availability *in vivo* due to the coupling of pyruvate-derived acetyl-CoA with microbial growth and energy metabolism via the oxidative tricarboxylic acid cycle. A genome-scale metabolic model combined with three algorithm analyses was used for strain design. Carbon metabolism was reprogrammed using two genetic control tools to fine-tune gene expression. Adaptive laboratory evolution and multi-omics analysis screened the growth-related regulatory targets beyond rational design. A novel metabolic pattern of glucose-grown bacteria is established to maintain growth fitness and minimize carbon loss under aerobic conditions for the synthesis of pyruvate-derived products. This study provides valuable insights into the design of a microbial cell factory for synthetic biology to produce industrial bio-products of interest.

**KEYWORDS** genome-scale metabolic model, cistronic library, T7 RNAP-mediated sRNA, pyruvate, adaptive laboratory evolution, multi-omics analysis

Address correspondence to Yun Zhang, zhangyun@im.ac.cn, or Tingyi Wen, wenty@im.ac.cn.

Yun Zhang and Xueliang Wang contributed equally to this article. The Author order was determined by their contribution to the article.

The authors declare no conflict of interest.

See the funding table on p. 25.

Bio-manufacturing industrial chemicals from biomass-derived low-cost carbon sources using microbes as a renewable cell factory provides an eco-friendly alternative to current petrochemical-based processes (1, 2). To re-routing carbon flux to achieve a desired production phenotype, the genome-scale metabolic model as an important technological tool for understanding cellular metabolism is combined with constraint-based algorithms for rational strain design (3, 4), and efficient genomic editing tools have been developed to accelerate the process of metabolic engineering (5–8). The severe resource competition between biomass and bioproduction needs multiple rounds of the "design-build-test-learn" cycle to reprogram the complicated metabolic network for production enhancement (9).

Pyruvate serves as a central metabolite to bridge the glycolytic pathway and tricarboxylic acid (TCA) cycle and is a crucial building block for the biosynthesis of amino acids and industrial chemicals such as ethanol, 2,3-butanediol, isobutanol, and isopropanol (2, 10, 11). During aerobic growth on glucose, pyruvate is oxidized by pyruvate dehydrogenase complex (PDHC) to generate acetyl-CoA fueling the TCA cycle for energy production, causing up to 50% loss of pyruvate via carbon dioxide (12). Another part of pyruvate is recruited to synthesize amino acids for maintaining growth, so metabolic flux at the pyruvate node is flexible (13). The strategy for engineering two industrial platform microorganisms, *Escherichia coli* and *Corynebacterium glutamicum,* to improve pyruvate availability depends on minimizing the carbon flow toward pyruvate-depleting pathways mainly by blocking pyruvate oxidation (14, 15). Whether limiting lipoic acid (a required cofactor) to decrease PDHC activity or deleting *aceE* encoding the E1p subunit of PDHC to inhibit the TCA cycle causes the severe growth defect on glucose and generates a lipoic acid or acetate auxotroph strain (16–18). Modulation of the stringent response (ppGpp), the introduction of an ATP futile cycle system, and the disruption of a global regulator RamA have been attempted to improve pyruvate accumulation without growth retardant (18–20). The challenge of engineering a novel platform strain for pyruvate-derived product synthesis is to create a metabolic scenario whereby the bioproduction process is adapted to cellular growth, achieving metabolic homeostasis between biomass formation and product synthesis (21).

To overcome the competing task in bioprocess, many engineering tools are developed to achieve varied levels of gene expression for optimizing metabolic flux distribution. Gene expression levels are normally determined by many factors, such as promoters, ribosome-binding sites (RBS), 5′ untranslated region (5′ UTR), and translation initiation region (22). Promoter and RBS are engineered to enhance gene expression (23, 24); however, these elements might not function as expected with different coding sequences due to the effect of differential mRNA secondary structures on translation initiation rate. To eliminate the inhibitory stem-loop structure of mRNA, a bicistronic design (BCD) with a sequence-economic translated polypeptide can regulate gene expression level by translation coupling to a leader peptide as a first cistron with variant translation initiation rate (25). Despite BCD is convenient to overexpress non-inherent heterologous genes for the construction of new genetic circuits, the leader cistronic elements are deficient for predictable forward engineering of gene expression over a wide dynamic range. As for a multiple-target metabolic network, RNA-based regulatory toolkits are applicable for the simultaneous modulation of many gene expressions at the genome-scale level (26–28). The synthetic small RNA (sRNA) has been engineered for gene knockdown in a tunable manner by binding a target mRNA to prevent the entry of 30S ribosome and block the translation process (29, 30). The repression efficiency of sRNA could be modulated by optimizing the target-binding sequence, modifying abundances by promoters with different strengths, and broadening the spectrum range with sRNA scaffold engineering (31–34). As for downregulating essential genes with a high-level expression, it is difficult to produce abundant sRNA at a short-time response for metabolic flux redirection. Desired traits could not simply be engineered by rational design due to the overwhelming complexity of biological systems; thus, adaptive laboratory evolution (ALE) can mimic the natural evolution process to screen the strain

with improved properties (35–37). Under an artificial selective pressure, ALE can rapidly generate desired phenotypes by enriching mutations on metabolic enzymes, rewiring serendipitous pathways, and changing transcriptional profiles, resulting in the fitness advantage of growth rate or product yield/titer (38–40).

In this study, model-guided central carbon metabolic reprogramming was performed using efficient genetic tools and ALE for improving pyruvate availability to synthesize L-leucine in *C. glutamicum*. A library of ~3,000 leader cistronic variants was constructed to fine-tune the expression of the target gene at the translation stage. Using cistronic elements to express heterologous *fxpK* in a bicistronic cassette, a bifido shunt pathway was introduced to bypass pyruvate oxidation for acetyl-CoA synthesis. The dynamic control by T7 RNAP-mediated synthetic sRNA incurred pyruvate overflowing from cell growth toward product synthesis. ALE rescued cell growth through flux redistribution at the isocitrate branch by activating the glyoxylate shunt and attenuating the TCA cycle with a mutated isocitrate dehydrogenase as a metabolic switch. The metabolic rewiring enhanced the carbon utilization efficiency under aerobic conditions and achieved a high titer of L-leucine, which provides new strategies for engineering strain to synthesize pyruvate derivatives of industrial interest.

## RESULTS

### *In silico* design and reprogramming pyruvate metabolism

A genome-scale metabolic model-based strategy was employed to identify genetic targets for strain design to produce pyruvate-derived L-alanine, L-valine, and L-leucine (Fig. 1A). The simulation by the *i*CW773 model of *C. glutamicum* showed that the synthesis of L-alanine, L-valine, and L-leucine competed with biomass formation for carbon flux (Fig. S1), and the theoretical maximum yield from glucose is 1.97 mol/mol for L-alanine, 1.00 mol/mol for L-valine, and 0.66 mol/mol for L-leucine, respectively (Table S1). Based on the 20% lowest biomass constraint, three algorithm analyses [flux balance analysis (FBA), geometricFBA (gFBA), and flux variable analysis (FVA)] showed that *ldh* and *alaT*-catalyzed reactions were unessential and substitutable (Fig. 1A) and could be blocked to decrease pyruvate depletion. A common observation was the metabolic shift of pyruvate from entering the TCA cycle to flow into target products (Fig. S2; Table S2), indicating that *aceE*, *gltA*, *acn,* and *icd* genes essential for cell growth need to be downregulated to redirect pyruvate flux. To replenish oxaloacetate for the TCA cycle, two C4 anaplerotic pathways (*pyc*-encoding pyruvate carboxylase and *ppc*-encoding phosphoenolpyruvate carboxylase) can generate oxaloacetate from pyruvate and PEP (41). The defects in two C4 anaplerotic pathways abolished the aerobic growth of *C. glutamicum* on glucose (42). The deficiency of *ppc* gene had no effect on biomass or product synthesis, owing to Pyc as the principal C4 anaplerotic route responsible for about 90% of the total oxaloacetate synthesis (43). As assigning *pyc* flux to zero, *ppc* flux was significantly increased for biomass formation, and flux toward the TCA cycle decreased slightly (Table S3), which is consistent with the observation that the deficiency of Pyc incurred the lower accumulation of TCA cycle-derived amino acids (44). Therefore, the increase of pyruvate availability by blocking the unessential, substitutable reactions, and partial C4 anaplerotic pathway together with attenuating TCA cycle contributes to enhancing the biomass-coupled production of pyruvate-derived products.

To demonstrate metabolic strategies for increasing pyruvate availability for producing L-leucine as the target product, we first modified the L-leucine synthetic pathway by releasing the transcriptional repression of *leuCDB* genes by LtbR and the feedback inhibition of LeuA by L-leucine (Fig. 1B). The strong $P_{tuf}$ replacement resulted in 9-, 22-, and 30-fold increases in the transcriptional levels of *leuC*, *leuD,* and *leuB* genes and the inactivation of LtbR (Fig. S3A). As the inducible overexpression of the feedback-resistant *leuA*$^r$, the LEU-1 strain accumulated 3.44 ± 0.25 g/L L-leucine. Then, *ilvA* was deleted to decrease the split-flow at the *ilvBN*-catalyzed reaction, resulting in a decrease in the growth and glucose consumption rates of the LEU-2 strain. The supplement of trace isoleucine restored the growth of the LEU-2 strain, and the L-leucine titer improved to

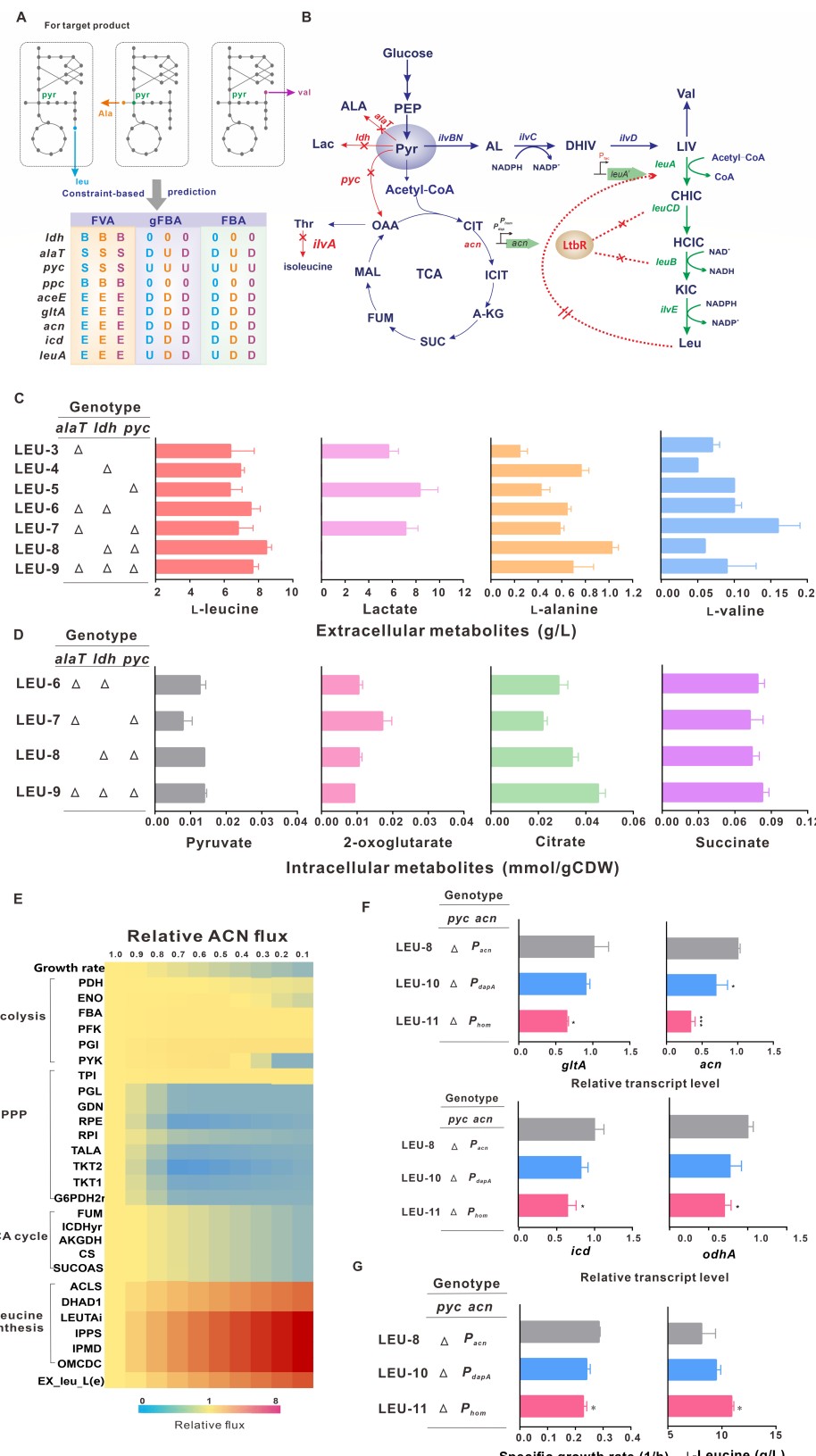

**FIG 1** *In silico* design and reprogramming pyruvate metabolism for ʟ-leucine accumulation. (A) The strategy predicted by FBA, gFBA, and FVA for engineering ʟ-leucine, ʟ-alanine, and ʟ-valine producers. The reaction essentiality is represented by E (essential), S (substitutable), and B (blocked), respectively. The regulation mode is represented by U (upregulation) and D (Continued on next page)

**FIG 1** (Continued)

(downregulation). (B) Scheme of the engineered metabolic pathway for L-leucine accumulation. The red X indicates gene deletion, and the green arrows represent gene overexpression. (C) The titers of L-leucine, L-valine, lactate, and L-alanine in shake flask cultivation. (D) The concentrations of organic acids in shake flask cultivation. (E) Simulation of the impact of decreased ACN flux on cellular metabolic flux distribution. Relative ACN flux = 1.0 served as the control (or native ACN flux). Yellow, red, and blue represent no change, upregulation, and downregulation of the flux. (F) Relative transcript levels of genes at the exponential growth phase. (G) The specific growth rate and L-leucine titer in shake flask cultivation. Data shown are mean values from three biological replicates and the standard deviations are presented. Significant differences in the data were determined using Student's $t$ test (*$P < 0.05$ and **$P < 0.01$).

5.86 ± 0.70 g/L (Fig. S3B). To decrease the split-flow of pyruvate, *alaT*, *ldhA*, and *pyc* were deleted separately, and the L-leucine titer increased by 9%, 19%, and 8%, respectively (Fig. 1C; Table S4). As expected, L-leucine titer enhanced with the paired combinatorial deletion of three genes and maximally improved to 8.46 ± 0.30 g/L in response to the deficiency of *ldh* and *pyc*. However, the deletion of *alaT*, *ldh*, and *pyc* combination showed a marginal effect on the intracellular pyruvate pool. The increase in the citrate pool indicated that pyruvate overflowed into the TCA cycle, blocking the bypass flow (Fig. 1D). We previously observed that aconitase acted as a major control node in the TCA cycle (45). *In silico* analysis showed that the flux toward L-leucine synthesis elevated with the decline of relative Acn flux (Fig. 1E). Two weak $P_{dapA}$ and $P_{hom}$ were inserted into the genome to replace the original promoter, resulting in 33% and 66% decreases in the mRNA level of *acn* in LEU-10 and LEU-11 strains, respectively. Meanwhile, a similar decreased trend on the mRNA levels of *gltA*, *icd*, and *odhA* genes was observed (Fig. 1F). The L-leucine titer increased by 10% and 26%; however, the specific growth rates of LEU-10 and LEU-11 strains decreased by about 20% (Fig. 1G), indicating that attenuating the TCA cycle decreases pyruvate split-flow at the expense of growth.

## A library of leader cistronic elements for translation modulation

To optimize flux distribution by modulating gene expression at the translation level, we constructed a library of cistronic elements to eliminate the effect of uncertainty arising from the 5′ UTRs on translation initiation rate through translation coupling in *C. glutamicum*. In the genetic architecture of BCD, the independently translated coding frame of the 41-nt leader cistron containing the RBS2 sequence was positioned upstream and overlapped with the translation initiation site of the downstream target gene, allowing a high efficiency to mediate the translation of the target gene (Fig. 2A). The coding sequences of 10 high-abundance proteins being screened from the proteomic profiling of *C. glutamicum* acted as the first leader cistron to couple with green fluorescent protein (GFP) translation, using the BCD2 (a standard and reliable translation element from *E. coli*) as a control. We observed that BCD2 resulted in a low level of fluorescence intensity in *C. glutamicum*, and Tsf as the leader cistron produced a very high-level fluorescence intensity. The GFP synthesis was sensitive to the changes in the coding sequences of the leader cistron (Fig. 2B). The GFP intensity was correlated with the folding free energy of the leader cistron ($\Delta G_{folding}$) rather than the hybridization energy ($\Delta G_{hybridization}$) of the leader cistron-16S rRNA duplex in the presence of the same RBS (Fig. 2B).

To evaluate the effect of sequence changes in the leader cistron on translation efficiency, the mutant library for the three regions of Tsf leader cistron (M1, M2, and M3) was constructed by installing degenerate bases via chemical synthesis, and the first round of screening 2,880 colonies was performed in 96-deep-well plates (Fig. 2C). The mutant library covered 1- to 76-fold variation range of fluorescence intensity. The M1 mutation produced maximal variants with a 2.5-fold increase in the absolute expression of GFP, and many variants in M3 region incurred a significant decrease in fluorescence intensity (Fig. 2D). An analysis of variance of observed fluorescence intensity showed that 44% of the total variants incurred a 50%–66% decrease in the absolute expression of GFP,

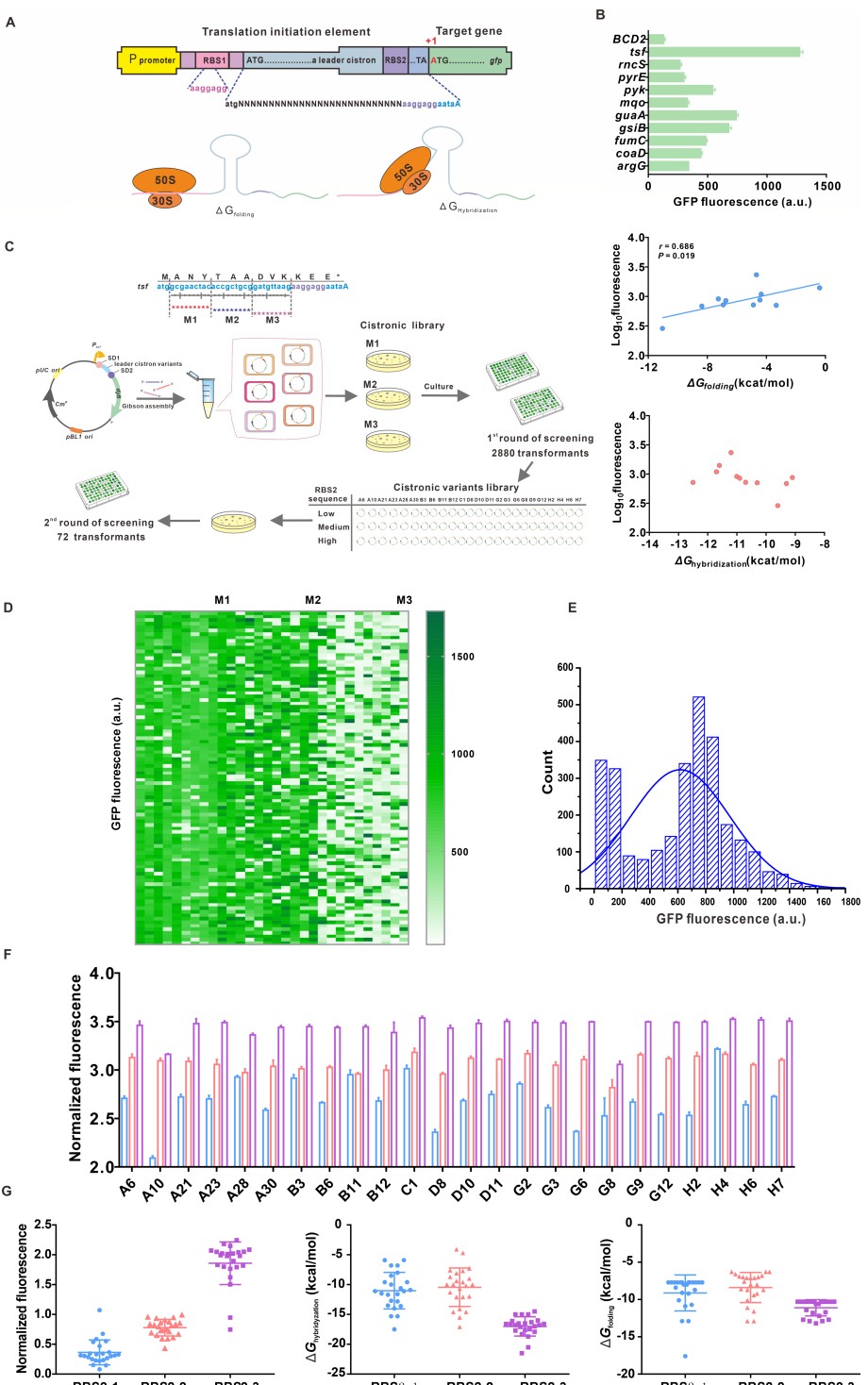

**FIG 2** A library of leader cistronic elements for translation modulation in *C. glutamicum*. (A) The genetic architecture of BCD with two RBSs to express the target gene. (B) GFP fluorescence, free energy of mRNA folding ($\Delta G_{folding}$), and hybridization energy of the leader cistron-16s RNA duplex ($\Delta G_{hybridization}$) for 10 leader cistrons. (C) A flowchart of constructing and screening the cistronic library. (D) GFP fluorescence intensities of 2,880 cistronic variants in 96-well microplates. (E) Distribution of GFP fluorescence intensity in the cistronic library. (F) Normalized GFP fluorescence intensities for the combination of 24 cistronic variants and three RBS2 sequences. RBS2-1, RBS2-2, and RBS2-3 with a low, medium, and strong strengths are shown in blue, pink, and purple columns, respectively. (G) Normalized GFP fluorescence intensity predicted $\Delta G_{folding}$, and $\Delta G_{hybridization}$ for three RBS2 groups.

and only 3.6% of the total variants produced a higher fluorescence intensity than the parent leader cistron (Fig. 2E), indicating that the change of leader cistron sequence can lead to the variance of translation coupling efficiency.

Considering that the RBS2 sequence initiates the translation process of the downstream gene in a BCD and is responsible for the recruitment of ribosomal RNA, three RBSs with low, medium, and high strengths (RBS2-1, RBS2-2, and RBS2-3) were assembled with 24 cistronic variants to test the influence of RBS2 strength on translation efficiency. A total of 72 transformants of the cistronic variant library were used for the second round of screening in 96-deep-well plates (Fig. 2C). All combinations of variants with RBS2-3 of strong strength caused 1.5- to 5.0-fold increases in the fluorescence intensity (Fig. 2F). A common observation was the correlation between the normalized fluorescence intensity and RBS2 strength (Fig. 2G). The hybridization energy in the RBS2-3 group was significantly lower than those in the RBS2-1 and RBS2-2 groups, demonstrating that the increase in the binding strength between the leader cistron and 16S rRNA contributes to producing a high abundance of protein. Remarkably, the combination of variants with RBS2-1 of low strength showed a significant difference in fluorescence intensity, indicating that the translation efficiency of the second cistron might be sensitive to the sequence change of the leader cistron in response to RBS2 of low strength. These characterized leader cistrons with ~160-fold range of reporter-protein expression provide available regulatory elements to control the translation efficiency of the target gene.

## Constructing a bifido shunt pathway using a cistronic library to increase non-pyruvate-derived acetyl-CoA supply

To bypass pyruvate oxidation, we tried to introduce a non-pyruvate-derived acetyl-CoA synthetic pathway for fueling the TCA cycle. The "bifido shunt" can partially bypass pyruvate through a phosphoketolase pathway in bifidobacteria (46, 47). A bifunctional phosphoketolase (FxpK) from *Bifidobacterium adolescentis* catalyzes the conversion of fructose-6-phosphate (F6P) to erythrose-4-phosphate (E4P) and acetyl-phosphate (AcP), and xylulose-5-phosphate (X5P) to glyceraldehyde-3-phosphate (G3P) and AcP (Fig. 3A). Meanwhile, E4P and G3P can be converted into AcP via a series of carbon atom rearrangements, and AcP can be converted to acetyl-CoA by *pta*-encoding phosphate acetyltransferase, finally generating three acetyl-CoA molecules per glucose without carbon loss (Fig. 3A). When adding the FxpK-catalyzed reaction to *i*CW773, the acetyl-CoA from the bifido shunt fueled the TCA cycle to maintain biomass formation, and L-leucine yield increased from 0.58 to 0.63 mol/mol (Fig. 3A; Table S5).

To construct a bifido shunt pathway, the codon-optimized *fxpK* under the control of the strong constitutive $P_{tuf}$ was co-expressed with *leuA$^r$* mediated by the constitutive $P_{glyA}$ (Fig. 3B). The function of Fxpk was confirmed by detecting the amount of intracellular AcP. Both LEU-12 and LEU-13 strains generated AcP of 0.044 and 0.049 mmol/gCDW (Fig. 3C). The mRNA levels of *zwf*, *tkt*, *tal,* and *pta* genes were increased in response to FxpK expression (Fig. S4). Consequently, FxpK expression restored the specific growth rate and improved L-leucine titer to 11.73 ± 0.63 g/L (Fig. 3D). Remarkably, the total yield of biomass and L-leucine from glucose increased by 17% and 16% in LEU-12 and LEU-13 strains (Table S4), indicating that the introduction of bifido shunt pathway significantly increases carbon usage efficiency by decreasing carbon dioxide generation.

The carbon split ratio into the glycolysis and bifido shunt pathway depends on the abundance of FxpK; therefore, five leader cistron variants [*tsf(D10)*, *tsf(G6)*, *tsf(G9)*, *tsf(H4),* and *tsf(H7)* containing RBS2-3 sequence] were used to increase heterologous *fxpK* abundance by translation coupling. As expected, the specific growth rate, L-leucine titer, and yield increased in the corresponding strains (Fig. 3E; Table S4). The LEU-17 strain harboring a $P_{tuf}$-*tsf(H4)-RBS2-3-fxpK* bicistronic cassette showed the highest increase in L-leucine titer compared with the LEU-13 strain.

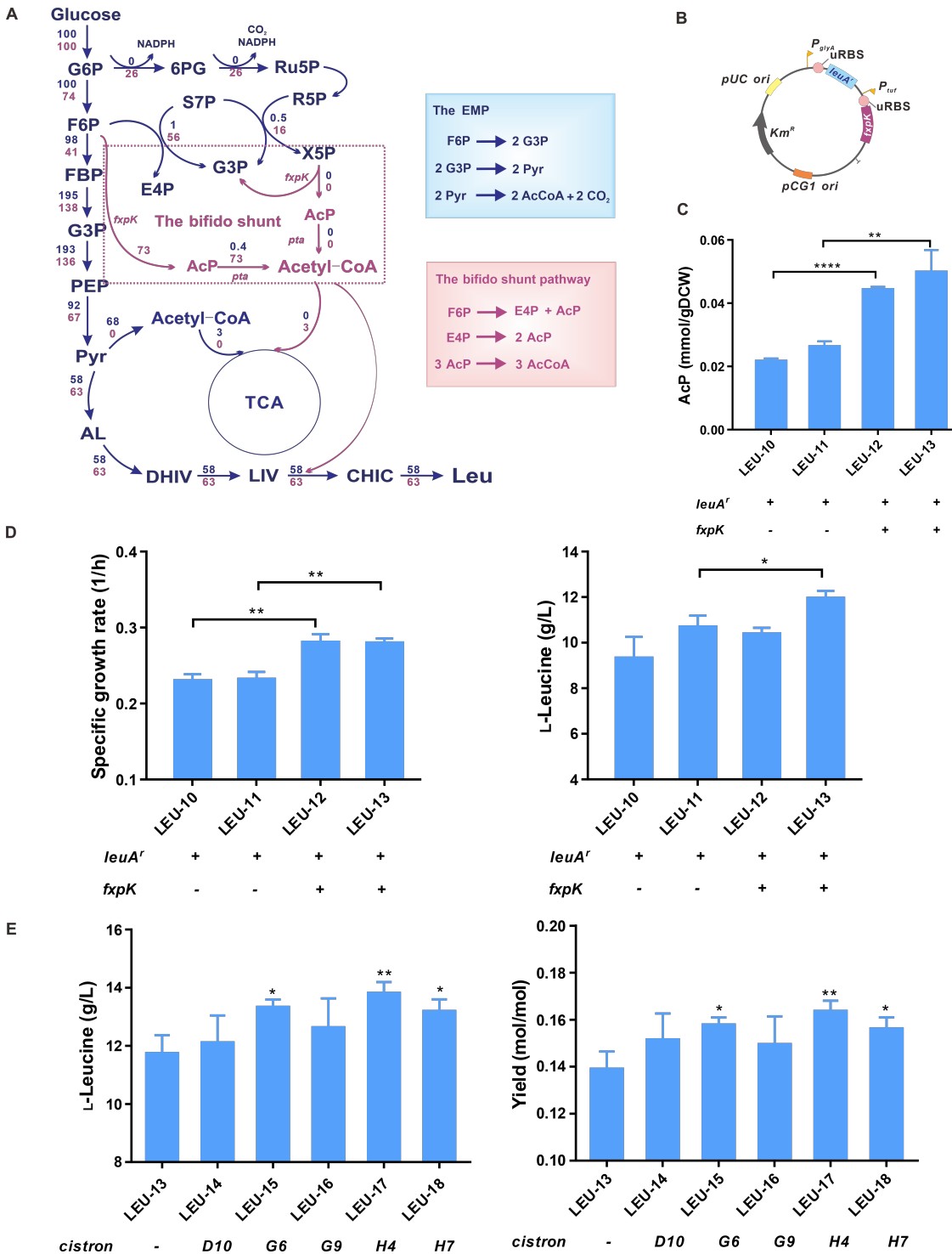

**FIG 3** Construction of a bifido shunt pathway using a cistronic library. (A) *In silico* simulation of flux distribution in response to the introduction of bifido shunt pathway. (B) The genetic map of p*leuA*ʳ*fxpK*. (C) The activity of *fxpK*-encoding phosphoketolase determined by a colorimetric assay to quantify the amount of AcP. (D) The specific growth rate and ʟ-leucine titer of LEU strains in shake flask cultivations. (E) The ʟ-leucine titers and yields of LEU strains harboring p*leuA*ʳ-P*tuf*-*tsf(M)*-*RBS2-3-fxpK* derivatives. Data shown are mean values from three biological replicates and the standard deviations are presented. Significant differences in the data were determined using Student's *t* test (*$P < 0.05$ and **$P < 0.01$).

## Dynamic downregulation of the TCA cycle via a T7 RNAP-mediated sRNA switch

To knockdown essential genes with a high-level expression for metabolic flux redirection, we developed sRNA system that was independent on host's specific promoters and RNA polymerase and was specifically recognized and executed by bacteriophage T7 RNA polymerase (T7 RNAP) (Fig. 4A). Based on the genetic architecture of synthetic sRNA, three synthetic sRNA were designed with a *gfp*-target-binding sequence combined with different scaffold sequences derived from MicC, DsrA, and MicF, respectively (Fig. 4B). The *gfp*-target-binding sequence spanned the region from −4 to +48 nt of *gfp,* and marginal residues were mutated to maintain the stem structure with a lower folding energy (Fig. 4B). We compared the repression efficiency of $P_{T7}$- and $P_{tac}$-mediated sRNA by targeting $P_{J23119}$-mediated *gfp* on an all-in-one plasmid in *E. coli*. Anti-*gfp* sRNA-1, sRNA-2, and sRNA-3 led to a decrease in fluorescence intensity, and the repression efficiencies of $P_{T7}$-mediated sRNA group were significantly higher than those of $P_{tac}$-mediated sRNA group under the same induction condition (Fig. 4C), demonstrating the effectiveness of $P_{T7}$-mediated sRNA system. Meanwhile, MicC as a scaffold showed an increase in the repression efficiency (Fig. 4D).

Two regulatory modes were designed to broaden the dynamic range of the T7 RNAP-mediated sRNA system. One design was based on T7 promoter variants to regulate the expression level of sRNA (Fig. 4E). Six $P_{T7}$ variants with strengths ranging from 81% to 166% relative to T7 promoter were used to mediate sRNA transcription. The repression efficiency of $P_{T7(H9)}$-mediated anti-*gfp* sRNA-1 increased to 86%, which was 30% higher than that of $P_{T7(4A6)}$-mediated anti-*gfp* sRNA-1. The repression efficiency of anti-*gfp* sRNA was positively correlated with the strength of T7 variants (Fig. 4E). In the other regulatory mode, the level of T7 RNAP expression was regulated by RBS of different strength (Fig. 4F). The repression efficiency of sRNA showed a positive correlation with the strength of RBS at different times of isopropyl-β-D-thiogalactoside (IPTG)-inducible expression of T7 RNAP. With the increasing inducible time, the repression efficiency of sRNA improved to above 90%, which was higher than that of $P_{T7}$ variants-mediated sRNA (Fig. 4F). It indicates that the increase in transcription machinery for sRNA contributes to improving the repression efficiency of sRNA. To further verify this effect, GFP expression under a strong $P_L$ promoter control was regulated by the sRNA system equipped with T7 RNAP of low or high abundance (Fig. S5A). As expected, the fluorescence intensity of $P_L$-mediated GFP was significantly higher than that of $P_{J23119}$-mediated GFP (Fig. S5B), and the repression efficiencies of anti-*gfp* sRNAs in the high-abundance T7 RNAP group still reached above 80% (Fig. S5C). Consistent with the observation in *E. coli*, the repression efficiencies of three anti-*gfp* sRNAs mediated by $P_{T7}$ were higher than those mediated by $P_{tac}$ in *C. glutamicum* (Fig. S6A). Unexpectedly, the introduction of Hfq showed little effect on the repression efficiencies of anti-*gfp* sRNAs (Fig. S6B). Three $P_{T7}$-mediated anti-*gfp* sRNAs incurred about a 70% decrease in GFP intensity with the increase of induction time (Fig. S6C).

T7 RNAP-mediated sRNA was used as a metabolic switch to dynamically control the expression of model-predicted *aceE* and *gltA* genes for the controllable distribution of pyruvate flux between cell growth and L-leucine biosynthesis (Fig. 5A). We constructed anti-*aceE* and anti-*gltA* sRNAs with $P_{T7(C4)}$ and $P_{T7(H9)}$ variants and a strong RBS to mediate the translation of T7 RNAP, respectively (Fig. 5B). In shake flask cultivations, cell growth was maintained at a comparable specific growth rate to the base strain for 12 h (Table S4), and then *anti-aceE* or *anti-gltA* sRNAs were induced to redirect the carbon flux, resulting in a 7% increase in L-leucine titer. Then, anti-*aceE* and anti-*gltA* sRNAs were co-expressed in configurations with different T7 variants and scaffolds (MicF or DsrA). The recombinant strains harboring two sRNAs showed different increases in L-leucine titer and yield (Fig. 5C). Consequently, the LEU-20 strain harboring $P_{T7(C4)}$-*anti-aceE* and $P_{T7(H9)}$-*anti-gltA* sRNAs produced 16.04 ± 0.50 g/L L-leucine with a yield of 0.22 mol/mol, which is 16% higher than that obtained without sRNA expression. The improvement in

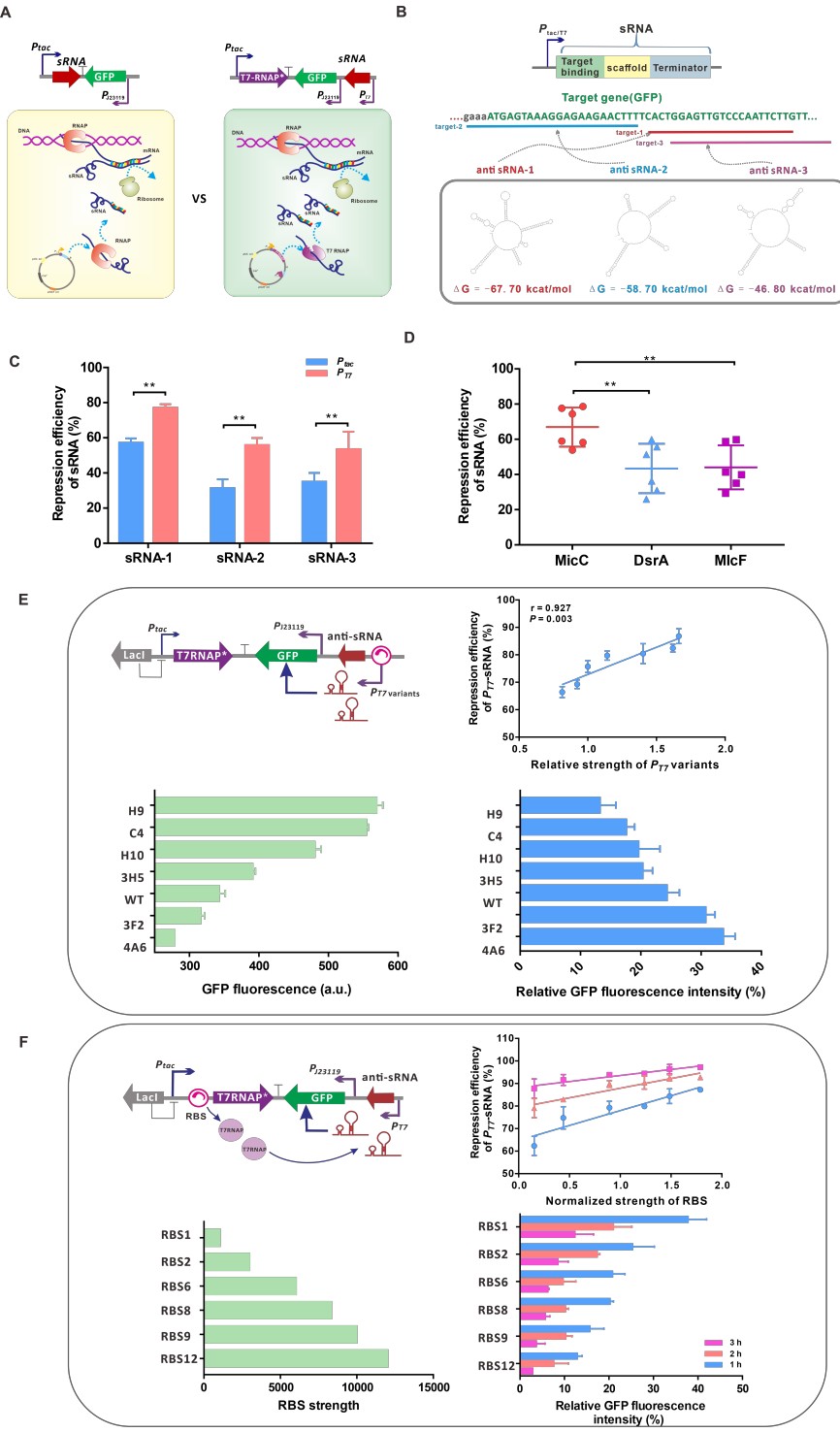

**FIG 4** A T7 RNAP-mediated sRNA system. (A) The comparison of $P_{tac}$ and $P_{T7}$-mediated anti-sRNA to knockdown gene expression. (B) The anti-sRNA consists of targeting binding residue, scaffold, and terminator. Twenty-four-nucleotide targeting residues and folding energies of anti-sRNA-1, sRNA-2, and sRNA-3 are shown in red, blue, and purple, respectively. (C) The repression efficiency of $P_{tac}$ and $P_{T7}$-mediated anti-sRNA to block GFP expression. (D) The repression efficiency of anti-sRNA with the sequences derived from MicC, DsrA, and MicF as a scaffold. (E) The relationship between the strengths of T7 promoter variants and the repression efficiency of anti-sRNA. (F) The relationship between the strengths of RBS-mediating T7 RNAP translation and the repression efficiency of anti-sRNA.

yield is attributed to the increased proportion of carbon fluxes recruited to L-leucine synthesis due to the dynamic control by the sRNA switch.

## Growth-driven adaptive laboratory evolution for carbon flux redistribution

To screen the growth-related regulatory targets beyond rational design, we abolished aerobic growth by blocking two C4 anaplerotic pathways in the LEU-28 strain to generate evolutionary pressure to rescue cell growth (Fig. 6A). ALE was conducted by serial passaging of initial strain into a glucose-minimal medium supplemented with yeast extract to support the initial growth and allow self-mutation for growth recovery (Fig. 6B). For every 7 days, yeast extract concentration gradually decreased to 0.8, 0.6, 0.4, 0.2, and 0 g/L, respectively (Fig. 6C). The biomass of evolved population had been increasing during every 7 days, indicating the occurrence of an evolutionary event (Fig. 6D). At the end of ALE, the continuous evolved strain could grow on glucose, suggesting that mutations beneficial to growth are fixed over time in the population (Fig. 6E). Four isolates from evolved populations from 0.6, 0.4, 0.2, and 0 g/L of yeast extract were transformed by p*leuA'-tsf(H4)-RBS2-3-fxpK* for shake flask cultivation. All the ALE strains showed a comparable growth rate and a higher biomass production than those strains harboring T7 RNAP-sRNA (Table S4). In particular, the L-leucine titer and yield in ALE0 strain amounted to those in the LEU strains harboring T7 RNAP-sRNA. The total yield of biomass and L-leucine from glucose in the ALE0 strain increased by 20% compared to that in the LEU-27 strain (Table S4), indicating that metabolic reprogramming during ALE improves carbon usage efficiency. In addition, morphological analysis and cell size determination observed the variance in cell length and shape (Fig. 6F). The average cell size of the LEU-28 strain was 26% longer than that of the WT strain, which was caused by blocking two C4 anaplerotic pathways. In contrast, the average cell size of the ALE0 strain decreased by 21% compared to the LEU-28 strain and was comparable to that of the WT strain (Fig. 6G), indicating that the accumulated mutations during ALE could restore cell length and growth when its variation was triggered.

Multi-omics analysis was performed to explore the changes in intracellular metabolism (Fig. 7A). Genomic DNAs of six ALE isolates were sequenced to identify the pool of mutations in the evolved strains. Thirty same mutations (nucleotide changes) occurred in the different evolved strains (Fig. S7; Table S6). Remarkably, all five isolates had mutations in the encoding and regulator regions of ABC-type transporters and a transcriptional regulator of sugar metabolism. Therefore, the growth performances of five isolates were further evaluated in the minimal medium with different carbon sources. Fructose and sucrose could not be catabolized by the evolved strains except for glucose and ribose (Fig. S8), indicating that these mutations have little effect on glucose transport. Moreover, a mutation in the *icd*-encoding monomeric isocitrate dehydrogenase (IDH) was shared with the evolved strains (from ALE0.8 to ALE0) (Fig. 7B), hinting that the redistribution of metabolic fluxes might be driven by changes in the TCA cycle. Comparative transcriptional analysis between ALE0 and LEU-28 strains showed that about 31% of 3,064 detected genes was differentially expressed. The numbers of significantly (1 > or −1 < $\log_2$ fold change with adjusted $P < 0.05$) up- and downregulated genes were 865 and 85 in the ALE0 strain compared with the parent strain (Fig. 7C; Table S7). As in the case of intracellular targeted metabolomics, there was a significant decrease in 51 metabolite pools and a distinct increase in 54 metabolite pools (Fig. 7D; Table S8). Principle component analysis of transcriptomic and metabolomics data showed that the ALE0 strain was separated from the parental strain (Fig. S9). The gene expression profiles of ALE0 and LEU-28 strains were distinguished in metabolic pathways related to glycolysis, TCA cycle, pentose phosphate pathway (PPP), phenylalanine, valine, glycine, arginine, and histidine metabolism, as well as purine metabolism (Fig. 7E; Fig. S10A). Based on KEGG pathway annotations, the pool sizes of metabolites involved in carbohydrate, amino acid, and purine metabolism were significantly different in the ALE0 strain compared to the parental strain (Fig. 7F; Fig. S10B).

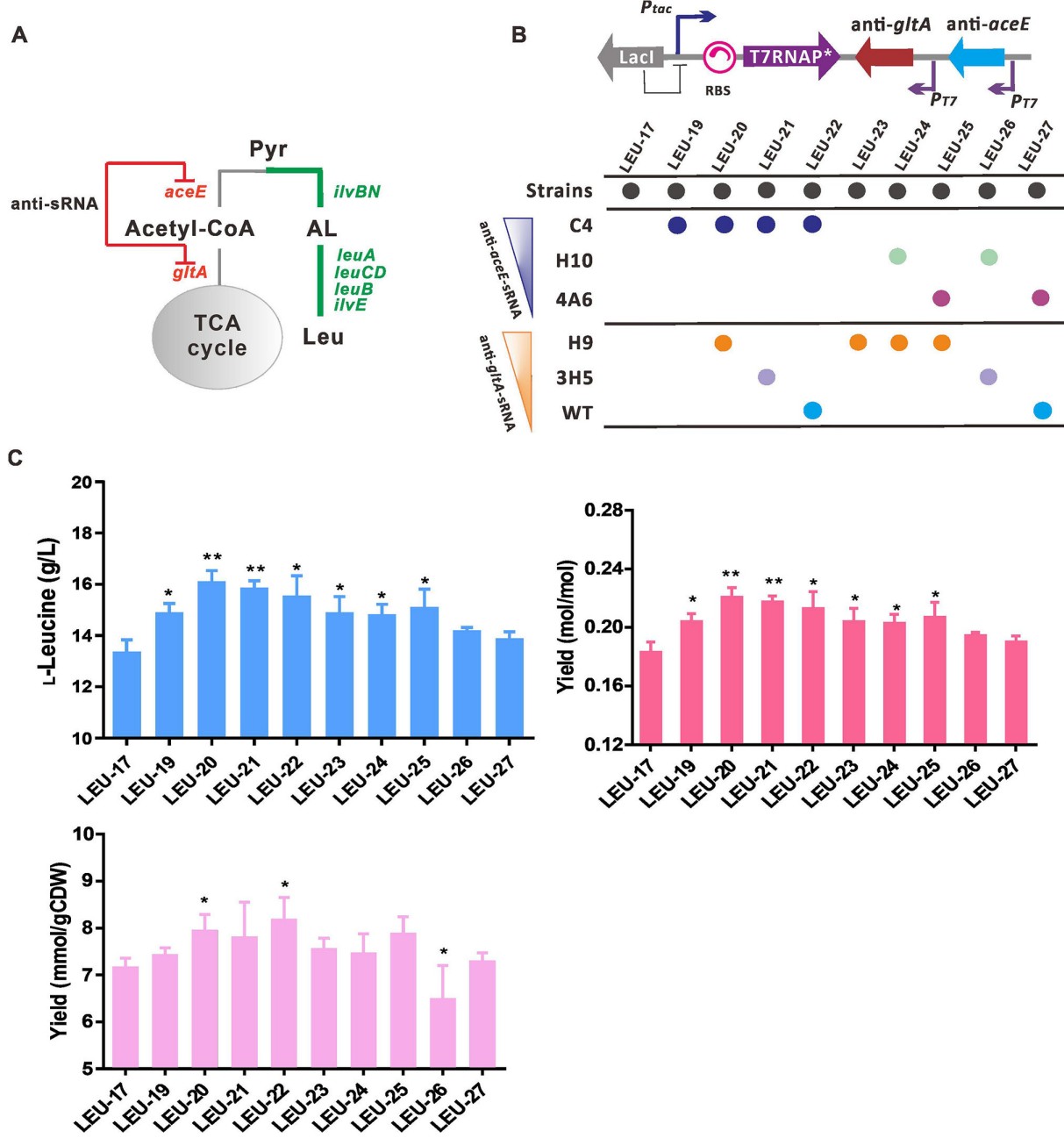

**FIG 5** Dynamic downregulation of the TCA cycle by a T7 RNAP-mediated sRNA switch. (A) Schemes of T7 RNAP-mediated sRNA for the dynamic control of *aceE* and *gltA* genes to improve pyruvate availability. (B) Engineered strains were constructed by combining and modulating the expression of anti-sRNAs with $P_{T7}$ variants to knockdown *aceE* and *gltA* genes. (C) The L-leucine titers and yields obtained for the engineered strains in shake flask cultivation. Data are presented as the means and standard deviations from three independent replicates. Significant differences were determined using Student's *t* test (*$P < 0.05$ and **$P < 0.01$).

The differences in transcriptome and metabolome profiling provided an insight into the change in cellular metabolic status owing to ALE. The genes involved in glycolysis were greatly upregulated, and the lower pools of metabolites derived from glucose were observed (Fig. 7E and F), indicating that glucose assimilation was enhanced in the ALE0 strain. The depletion of ribulose-5-phosphate occurred as a result of the upregulation of *zwf*, *opcA*, *pgl,* and *gnd* genes in PPP (Fig. S11), suggesting that more ribulose-5-phosphate was recruited to the synthesis of purine and aromatic amino acids in the ALE0 strain. Blocking two C4 anaplerotic pathways resulted in the accumulation of intracellular pyruvate and the lower pools of intermediates in the TCA cycle. In contrast, the

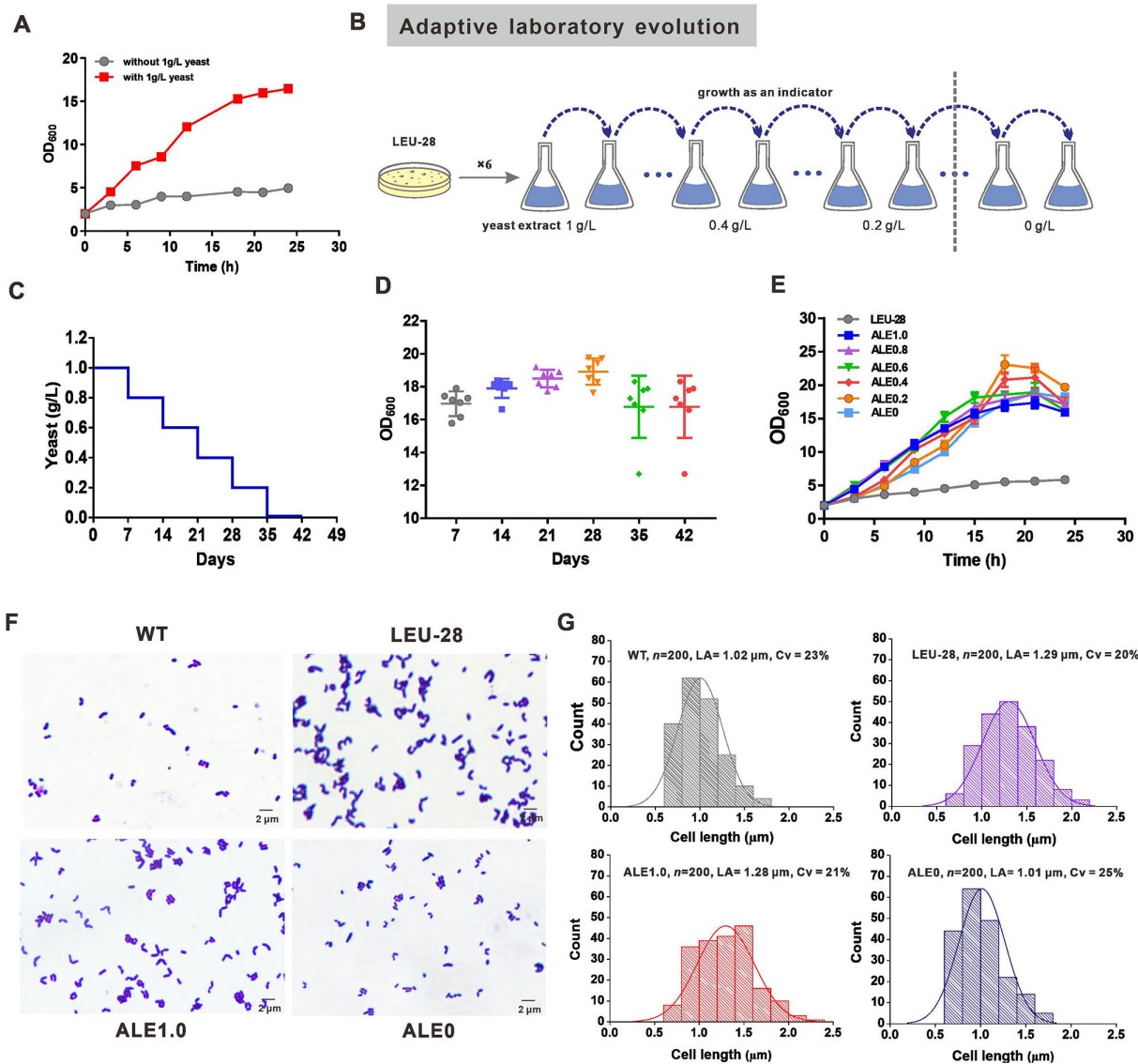

**FIG 6** Adaptive laboratory evolution of the LEU-28 strain. (A) Growth profile of the LEU-28 strain in the minimal medium in the presence and absence of 1 g/L of yeast extract. (B) The process of ALE using the glucose-minimal medium with various concentrations of yeast extract in shake flask cultivation. (C) Yeast extract supplementation was stepwise reduced from 1 to 0 g/L over time. (D) Cell growth trajectory showing changes in fitness during the ALE. (E) Growth profiles of one colony isolated from ALE in a minimal medium without yeast extract supplement. (F) Morphological observation of WT, LEU-28, ALE1.0, and ALE0 strains by microscope. (G) Cell size distribution in WT, LEU-28, ALE1.0, and ALE0 strains. $n$, number of analyzed individual cells; LA, average cell length; and $C_v$, coefficient of variation (%).

transcriptional levels of genes in the TCA cycle were elevated in the ALE0 strain, accompanied by a decrease in the acetyl-CoA pool (Fig. 7F). Notably, *aceA* and *aceB* in the glyoxylate cycle were significantly upregulated and the higher pool sizes of isocitric acid, fumaric acid, and malic acid were observed in the ALE0 strain (Fig. 7F; Fig. S11), indicating that the glyoxylate cycle was activated to replenish oxaloacetic acid for the recovery of cell growth. In addition, the ALE0 strain exhibited an increase in intracellular energy metabolite pools, including ADP and ATP (Fig. 7F), which was attributed to the upregulation of genes involved in electron transport chains (Fig. S11). Interestingly, genes related to ribosome protein, cell division, and peptidoglycan synthesis were upregulated in the evolved strain (Table S7). These findings reveal that global metabolic rewiring, specifically elevating glucose assimilation, activating the glyoxylate cycle, and enhancing

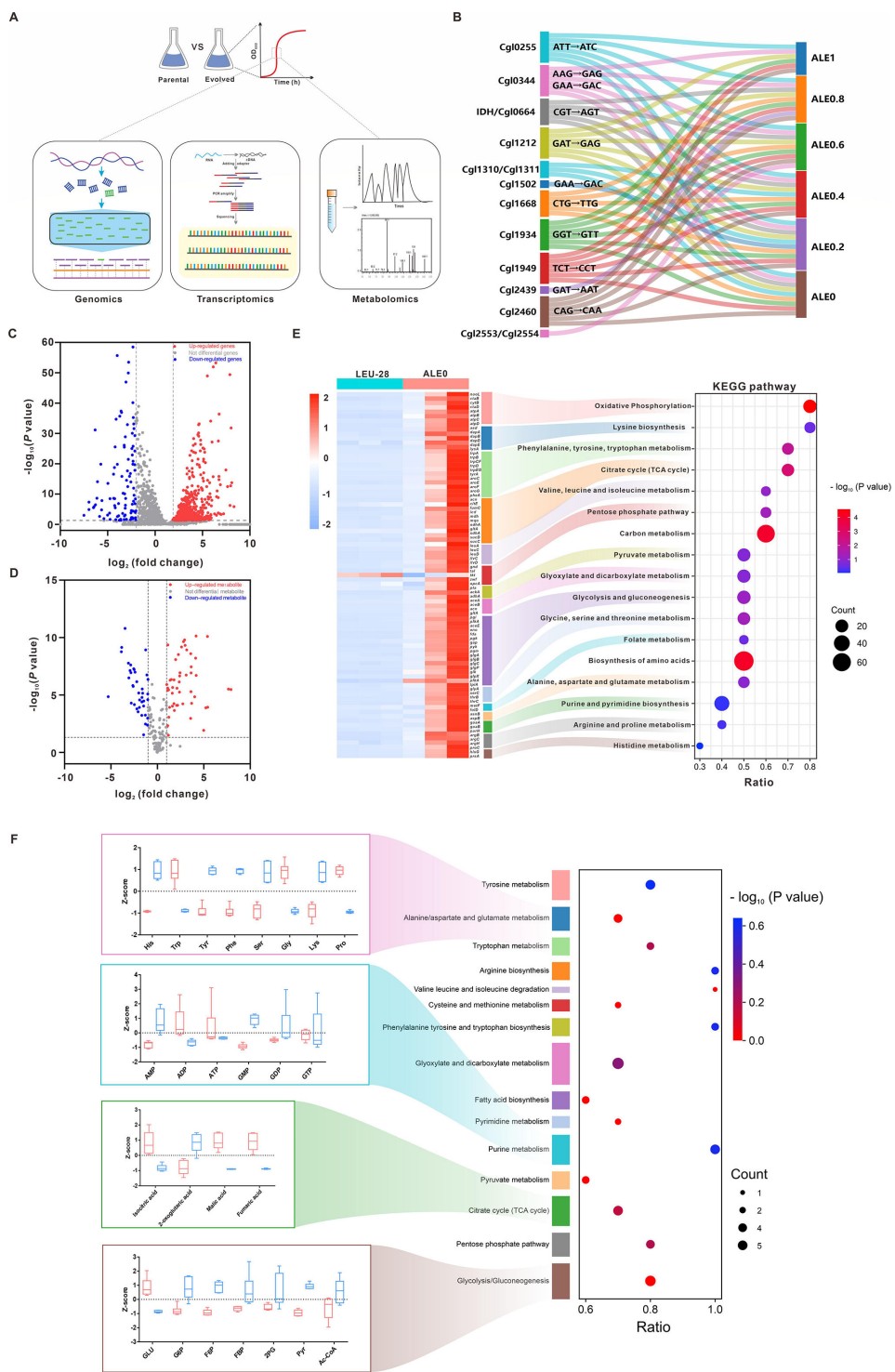

**FIG 7** Multi-omics characterization of the evolved strains. (A) Samples of the evolved and parental strains were collected at the mid-exponential growth phase for genomics, transcriptomics (*n* = 3 biological replicates), and targeted metabolomics (*n* = 6 biological replicates) characterization. (B) Mutations identified in the evolved population by whole-genome sequencing of six ALE strains (the Sankey diagram was built using the SankeyMATIC online tool). (C) Volcano plot showing the differentially expressed genes (DEG) identified by comparative transcriptomic analysis. (D) Volcano plot showing the differences in intracellular metabolite pools by targeted metabolomics analysis. (E) Heatmap and bubble plot for the enrichment of DEG. (F) Bubble plot for the enrichments of variant metabolite pools. *Z*-scores of detected metabolites are shown in column diagrams. Blue column represents the metabolite pools in the LEU-28 strain, and red column represents the metabolite pools in the ALE0 strain.

purine synthesis and oxidative phosphorylation, allows for the production of a certain amount of amino acids, nucleotides, and ATP to rescue cell growth.

## Evaluation and reverse engineering of the ALE-derived mutations

Notably, only a single nucleotide mutation (C to A) in all genomic mutations occurred in the *icd*-encoding IDH enzyme (R543S) involved in the TCA cycle. *In silico* docking calculation using the substrate-free model of IDH displayed the binding mode between isocitrate (ball and stick, colored by element) and IDH (Fig. 8A). Hydrogen bonds are formed between isocitrate and S130, N133, R137, R143, R543, Y416, and K253 residues of IDH (Fig. 8A). The monomeric IDH requires $Mg^{2+}$ to form the ion bonds with isocitrate; however, R543 is not involved in substrate binding. Furthermore, R543S mutation caused about an 80% decrease in IDH activity in ALE strains compared to the WT strain (Fig. 8B). The activation of glyoxylate shunt by omics analysis inspired us to detect the activities of isocitrate lyase (ICL) and malate synthase (MS). As expected, specific activities of ICL and MS in all ALE strains were significantly higher than those in the WT strain (Fig. 8B), demonstrating that IDH mutation resulted in the carbon flux redistribution between the TCA cycle and glyoxylate shunt.

To identify the contribution of mutated IDH for growth recovery, the mutation in *icd* gene was re-engineered into the initial LEU-28 strain. The resultant LEU-29 strain grew at the rate of 0.25 $h^{-1}$ in the minimal medium, with a 10-fold increase in biomass compared to the LEU-28 strain (Fig. 8C). As growth recovered, the L-leucine titer accounted for 48 folds higher than that of the LEU-28 strain, and the total yield increased by 11% (Table S4). Fed-batch fermentation was performed in a 5-L bioreactor to evaluate the L-leucine titer by the LEU-29 strain. The strain grew continuously with a maximal specific growth rate of 0.31 $h^{-1}$ and the biomass reached an $OD_{600}$ value of 125 (Fig. 8D). After 50 h cultivation, 17.92 g/L L-leucine was produced with a yield of 0.12 mol/mol glucose and a productivity of 0.36 g/L/h (Fig. 8D).

## A universal chassis for pyruvate-derived product synthesis by a controlled switch

In view of the findings by metabolic engineering and evolution, a novel metabolic pattern of glucose grown under aerobic conditions was established in a chassis strain to utilize the glyoxylate shunt as the only C4 anaplerotic pathway ($\Delta pyc\Delta ppcicd^{R453S}$) and the bifido shunt as the non-pyruvate-derived acetyl-CoA synthetic pathway [$::P_{tuf}$-*tsf(H4)*-*fxpK*] for minimizing carbon loss. A T7 RNAP regulatory module (*lacI*-$P_{tac}$-*T7RNAP*) was introduced into the genome of the chassis strain for the dynamic control of flux distribution to improve pyruvate availability. The potentials of chassis strain were evaluated for producing pyruvate-derived products using a controlled sRNA switch (Fig. 8E).

As for L-leucine synthesis, $P_{tac}$-mediated *leuA^r* together with two $P_{T7(C4)}$-anti-*aceE* and $P_{T7(H9)}$-anti-*gltA* sRNAs on an all-in-one plasmid were inducibly expressed in the chassis strain. In the fed-batch fermentation in a bioreactor, the specific growth rate was 0.32 $h^{-1}$ and biomass reached 30.24 CDW. The maximal L-leucine titer reached 22.46 g/L at 55 h with a yield of 0.16 mol/mol glucose and a productivity of 0.41 g/L/h (Fig. 8F). The L-leucine titer in this study is higher than that in the engineered strain overexpressing aminotransferase and is comparable to that in the engineered strain being deficient in two regulators, *ltbR* and *iolR* (48, 49). L-leucine precipitation occurred as the concentration exceeded above 23.7 g/L due to the low solubility in the water (49); therefore, the L-leucine titer in this study is close to the solubility of L-leucine. For L-alanine synthesis, the codon-optimized *alaD* encoding alanine dehydrogenase from *Lysinibacillus sphaericus* was overexpressed to construct a new synthetic pathway, and the sRNA switch was used for dynamic control of pyruvate flowing. In shake flask cultivations, the maximal L-alanine titer reached 27.62 ± 0.44 g/L with a yield of 0.70 mol/mol glucose (Fig. S12A). In the case of L-valine, the chassis strain harboring p*ilvBNCD* and sRNA switch accumulated 6.28 ± 0.27 g/L of L‑valine (Fig. S12B). The chassis developed in this study can be used as a host to synthesize pyruvate-derived products under aerobic conditions.

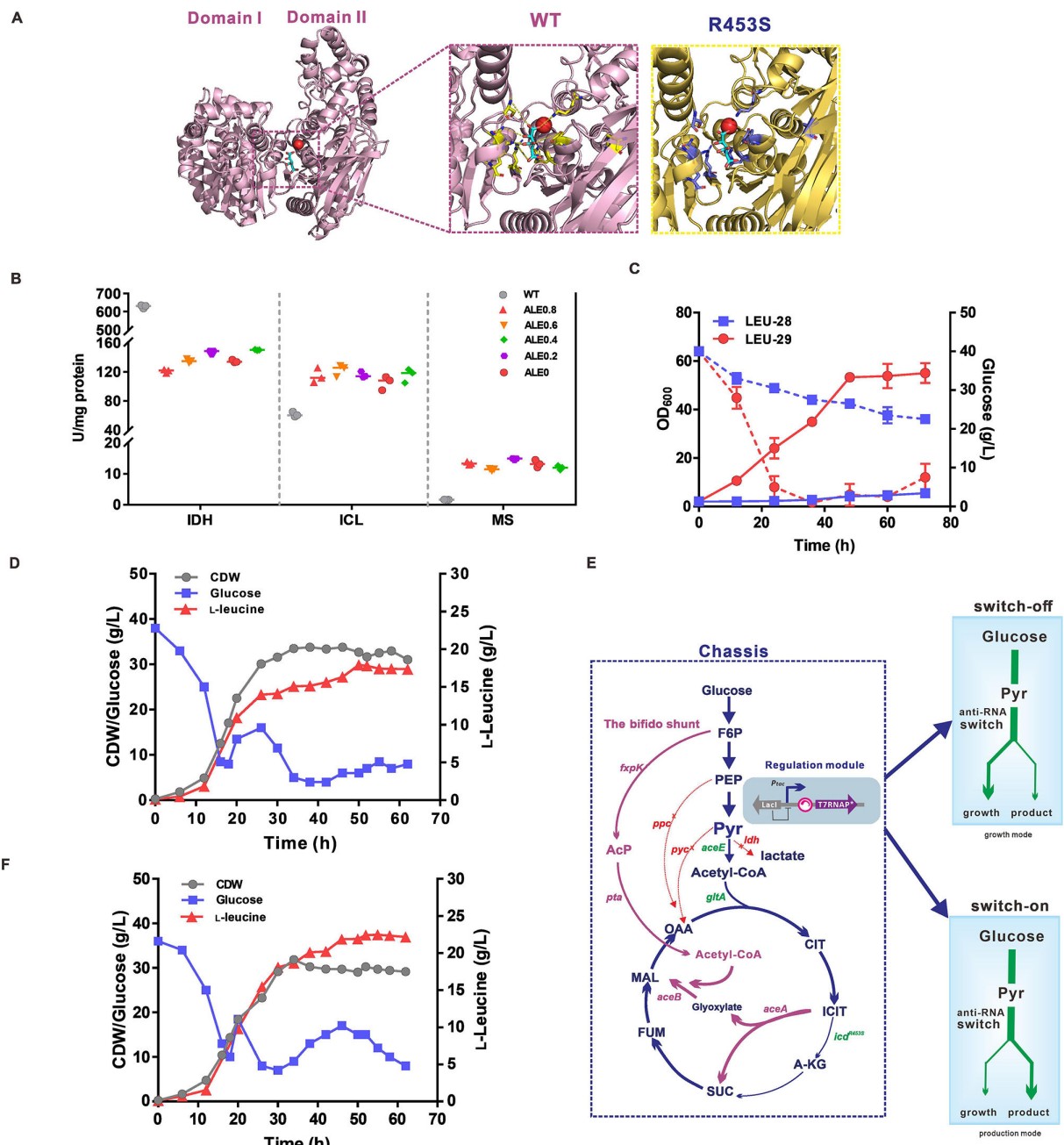

**FIG 8** Reverse engineering and construction of a chassis for producing pyruvate-derived products. (A) Docking of IDH with isocitrate in the binding pocket. Isocitrate and the residues involved in substrate binding in the model are represented as a ball-and-stick model. Magnesium ion is shown in the red sphere. The main hydrogen bonds are depicted as dashed yellow lines. (B) Specific enzyme activities of IDH, ICL, and MS in the ALE and parental strains. (C) Time profiles of cell growth and glucose concentration of the LEU-28 and LEU-29 strains in shake flask cultivation. (D) Fed-batch fermentation profile of the LEU-29 strain in a 5-L bioreactor. (E) Scheme of a chassis for producing pyruvate-derived derivative by a controlled switch. (F) Fed-batch fermentation profile of the engineered chassis strain for L-leucine production in a 5-L bioreactor.

## DISCUSSION

The quantitative regulation of metabolic flux toward the exogenous pathways or endogenous metabolism of genetic circuits depends on the change in gene expression level, which can be tuned by RBS of specified strength with a quantified promoter. RBS libraries now allow translational tuning of exogenous gene expression for pathway reconstruction or assembly of multiple genes for the inter-module balance (11, 50,

51). Despite machine learning algorithms having been developed to model the RBS sequence-phenotype relationship (52), the corresponding high-throughput screening methods were highly required to obtain the optimal RBS combinations (53). In addition, the translation efficiency of different genes was variable in response to the same RBS (54). A bicistronic design can alleviate the influence of target gene sequences on translation efficiency by coupling with a well-characterized leader cistron. We constructed and characterized a leader cistronic library, which was used to tune translation efficiency via translational coupling in BCD. When comparing the impacts of 10 leader cistrons on fluorescence intensity, we observe that translation efficiency mainly depends on the folding energy of the leader cistron, which is consistent with the observation in *E. coli* (55). Unexpectedly, BCD2 as a leader cistron produced a lower level of fluorescence intensity in *C. glutamicum* than that in *E. coli,* which might be attributed to the codon usage bias in the different species (56). To amplify the regulatory range of leader cistron on translation efficiency, nine nucleotide variance in three regions of Tsf leader cistron (M1, M2, and M3) can theoretically generate about $4^9$ mutants, which is lower than that arose from 18 nucleotide variance. It can not only cover a desired expression parameter space but also reduce library size and screening workload. Finally, we obtained the mutant cistronic library covering a 76-fold range. Moreover, the increase in RBS2 strength contributes to improving translation efficiency. The quantitative range of cistron-mediated translational coupling can be further expanded in combination with promoters of variable strength (25).

Pyruvate oxidation comprises the essential link between glycolysis and the TCA cycle and forms two molecules of carbon dioxide from one molecule of glucose (41). The PPP converts glucose to ribose 5-phosphate and unavoidably generates one molecule of carbon dioxide. These two pathways incur carbon loss by decarboxylation, thereby decreasing the carbon yield of the product from glucose. Two carbon-saving pathways, the non-oxidative glycolytic (NOG) and EP-bifido pathways had been engineered to decrease carbon loss (57, 58). The NOG pathway generating three molecules of acetyl-CoA from one molecule of glucose maximized the carbon yield without carbon dioxide loss but was unable to generate NADPH (57, 59). In spite of a high yield of acetyl-CoA from glucose by blocking glycolysis to divert F6P into bifido pathway, the EP-bifido pathway triggered the deficiency of phosphate donor PEP for PTS-mediated glucose transport and caused severe growth defect (58). The introduction of a PEP-independent transporter could restore the glucose uptake and growth (60), and overexpression of *zwf* was able to enhance PPP flux for NADPH supply in the presence of the EP-bifido pathway (61). As for L-leucine synthesis, we introduced the bifido shunt pathway without blocking the glycolysis pathway to bypass pyruvate oxidation, simultaneously meeting the demands of pyruvate supply for L-leucine synthesis and non-pyruvate-derived acetyl-CoA for fueling the TCA cycle. Consequently, the specific growth rate and biomass increased in response to the introduction of bifido shunt pathway. On the other hand, the elevated transcript levels of genes involved in PPP increased the carbon flux toward PPP to supply NADPH for L-leucine synthesis, supporting the model-based simulation. The bifido shunt pathway contributes to the increase in the total yield of L-leucine and biomass from glucose by reducing carbon loss by partially bypassing pyruvate oxidation.

Dynamic regulation implements "process control" at the cellular level to achieve autonomous distribution of cellular resources between growth and production as demand. The developed clustered regularly interspaced short palindromic repeats interference (CRISPRi) and sRNA technology enable it easy to investigate the effect of a large number of genes on growth and production. CRISPRi blocks multiple gene transcription using orthogonal sgRNAs in combination with dCas9 by obstructing the movement of RNA polymerase (62). However, dCas9 directly binds to several *E. coli* endogenous genes even in the absence of a single-guide RNA, thus notably affecting cell morphology and cell growth (63). dCas9 expression often incurs toxicity on cell viability; therefore, a low leakiness of controllable system is required for dCas9 expression. CRISPRi leads to a non-specific pleiotropic effect on neighboring genes

and is unavailable to modulate the expression of individual genes in a polycistron (64). Compared with CRISPRi, synthetic sRNA is capable of application in the regulation of essential genes indispensable for bacterial growth and multiple target gene knock-down (29). We developed a T7 RNAP-mediated sRNA dynamic control system that was independent of the host's machinery and able to switch off essential pathways in a reciprocal fashion. A major drawback in the application of synthetic sRNA is the requirement for the large size of well-characterized host-specific promoters to vary sRNA abundance (31, 32). The advantage of this system is that T7 RNAP directs sRNA transcription from a series of well-defined T7 variants that are tightly inactive in the absence of T7 RNAP. The transcription rate of T7 RNAP is eightfold faster than *E. coli* RNAP and can expose naked mRNA (65), which contributes to generating a high abundance of sRNA at a short-time response speed. sRNA transcription by T7 RNAP is orthogonal to host machinery and makes separate transcription machinery and source available for sRNA synthesis, thus reducing the competition for host RNAP to synthesize the essential gene for cell growth (66). T7 RNAP has been shown to function in a variety of hosts; therefore, the T7 RNAP-mediated sRNA toolbox achieves dynamic control of gene expression in strains with multiple backgrounds to facilitate the construction of an RNAi library at the genome-scale for metabolic network regulation.

The disturbance of cell metabolism might have some effects on cell morphology and physiological processes, such as cell shape, growth, and division (67, 68). The larger cell size of the LEU-28 strain indicated that the cells might be swollen by the decrease in the integrity of peptidoglycan. As reported previously, the deletion of *sucAB* gene incurred the deficiency of succinyl-CoA for the synthesis of diaminopimelate, an essential constituent of the cell wall, and perturbed cell wall formation (69). Similarly, blocking two C4 anaplerotic pathways resulted in oxaloacetic acid deficiency to incur the insufficiency of succinyl-CoA, thereby interfering cell wall synthesis. The average size of cells was restored in the ALE0 strain, accompanied by the recovery of growth capacity, indicating that the activated glyoxylate shunt replenishes oxaloacetic acid to fuel the TCA cycle for succinyl-CoA synthesis. In our previous study, C1 units' metabolism plays an important role in maintaining the peptidoglycan synthesis and cell shape of *C. glutamicum* (67). The pool of serine was low with an increase in the glycine pool in the ALE0 strain, indicating that upregulation of *glyA* converts more serine to generate C1 units for peptidoglycan synthesis. Therefore, the activated glyoxylate shunt and the elevated C1 units' metabolism contribute to maintaining the integrity of peptidoglycan and cell shape.

To rewire cellular metabolism using the growth rate as a screen marker, adaptive evolution incurred readjusted metabolism, thereby activating glyoxylate shunt to rescue cell growth. It attributed to the R453S mutation of *icd* gene in all evolved strains with the decreased IDH activity. Relative to monomeric IDH in *C. glutamicum*, the dimeric isocitrate dehydrogenase (ICD) was post-translationally modified through allosteric regulation in *E. coli* (70). The phosphorylation of ICD on the Ser113 residue by AceK causes electrostatic repulsion of the isocitrate, effectively inactivating the enzyme (71). Ser113 residue is hidden deep within a pocket and points away from the surface of Apo-ICD (72). However, the interaction between AceK and ICD triggers a conformational change of ICD, resulting in Ser113 adopting an exposed, outward-facing position to be more accessible for phosphotransfer (73). Dimeric ICD is regulated through the phosphorylation cycle catalyzed by AceK.

Different from ICD, the comparison of substrate- and NADP$^+$-binding IDH structures demonstrates that the binding of isocitrate or NADP$^+$ to IDH induces a domain shift, resulting in a more closed conformation at the active site (74). In a previous evolutionary event, A94D, R453C, and G407S mutation enabled IDH activity decrease, and these residues were also not located at the active site (40). We deduce that these mutated residues cause the change of IDH conformation or Ser substitution provides the phosphorylated site for allosteric regulation to interfere with isocitrate binding. The extent of ICD phosphorylation determined by AceK controls the carbon flux distribution between the TCA cycle and glyoxylate shunt in *E. coli* (70). It supports our findings that

the decreased IDH activity causes an activation of the glyoxylate shunt in *C. glutamicum* to metabolize isocitrate from glucose for cell growth. The R453S mutation in IDH presented in all isolated ALE strains, demonstrating that it is a beneficial mutation driving clonal expansion. Therefore, IDH acts as a metabolic switch for the flux partitioning in response to metabolic disturbance, thereby causing a metabolic shift from the TCA cycle toward the glyoxylate shunt.

The glyoxylate shunt bypasses the two decarboxylation steps in the TCA cycle to generate malate from two molecules of acetyl-CoA input, thereby providing carbon skeletons for biomass formation. It is off for the glucose-grown *C. glutamicum* and activated in the presence of acetate (75). The transcription of the *aceB-aceA* cluster in a divergent direction was regulated by a negative regulator RamB and a positive regulator RamA in *C. glutamicum* (76, 77). The upregulation of *aceBA* gene in the ALE0 strain demonstrated that the glyoxylate shunt was activated in spite of no mutation occurring in the intergenic region between *aceB* and *aceA* genes and two regulators. Notably, the pool of isocitrate was significantly higher in the ALE0 strain compared to the LEU-28 strain, indicating that the decreased activity of IDH incurs intracellular isocitrate accumulation. Owing to the lower affinity of ICL than IDH for isocitrate (78, 79), the pool size of isocitrate increased to a threshold sufficient to redirect flux through the glyoxylate shunt. Isocitrate might act as a signal molecule to increase the binding between RamA and the *aceBA* promoters to upregulate *aceBA* expression in response to the elevated isocitrate pool. Interestingly, the deficiency of *sucCD* activated the glyoxylate shunt as a partial bypass route for succinate formation with no changes in the transcriptional level of *aceBA* (80, 81). ICL activity is strictly controlled by allosteric regulation in response to the intermediates of the central metabolism (79). As a result, the decrease in the succinate pool due to *sucCD* deficiency led to the release of the inhibition of ICL by succinate to increase ICL activity instead of transcriptional activation. Therefore, this is not an all-or-none switch between the TCA cycle and glyoxylate shunt. The flux split ratio at the isocitrate branch point is governed by IDH and ICL activities in response to intracellular metabolite pools except for transcriptional control. We introduced mutated IDH into engineered *C. glutamicum*, thereby rewiring C4 metabolism by activating the glyoxylate shunt and attenuating the TCA cycle under aerobic conditions for glucose catabolism. The elevated flux toward glyoxylate shunt further reduces carbon loss, and the attenuated TCA cycle still provides the building blocks for growth, thereby making the catabolism of glucose in a more efficient way.

In summary, a novel metabolic pattern of glucose-grown bacteria is established with the glyoxylate shunt as the only C4 anaplerotic pathway and the bifido shunt as the non-pyruvate-derived acetyl-CoA synthetic pathway to maintain growth fitness and minimize carbon loss under aerobic conditions. The dynamic control by a T7 RNAP-sRNA system is available for modulating essential gene expression to balance growth and production. Our findings provide a paradigm for developing high-performance strains to bio-manufacture chemicals derived from pyruvate.

## MATERIALS AND METHODS

### Strains and cultivations

The strains used in this study are listed in Table S9. The *E. coli* strain EC135 was used as the cloning host (82). The wild-type *C. glutamicum* ATCC13032 was used as the initial strain for metabolic engineering. *E. coli* strains were cultured in Luria-Bertani medium at 37°C. *C. glutamicum* strains were cultured in brain heart infusion medium (BHI 37 g/L and sorbitol 91 g/L) at 30°C for competent cell preparation. If necessary, antibiotics were added at the following concentrations: 50 µg/mL of kanamycin or 34 µg/mL of chloramphenicol in *E. coli* for plasmid maintenance and 25 µg/mL of kanamycin and 10 µg/mL of chloramphenicol in *C. glutamicum* for recombinant screening.

The engineered *C. glutamicum* strains were cultivated in shake flasks for L-leucine production. Strains were precultured in the CGIII seed medium containing glucose 10 g/L, yeast extract 5 g/L, peptone 10 g/L, and NaCl 10 g/L at 30°C until the $OD_{600}$ reached 12 (83). One milliliter of seed culture was inoculated in a 500-mL baffled shake flask with 30 mL of CGX medium consisting of glucose 40 g/L, $(NH_4)_2SO_4$ 20 g/L, $KH_2PO_4$ 0.5 g/L, $K_2HPO_4 \cdot 3H_2O$ 0.5 g/L, $MgSO_4 \cdot 7H_2O$ 0.25 g/L, $FeSO_4 \cdot 7H_2O$ 0.01 g/L, $MnSO_4 \cdot H_2O$ 0.01 g/L, $ZnSO_4 \cdot 7H_2O$ 1 mg/L, $CuSO_4$ 0.2 mg/L, $NiCl_2 \cdot 6H_2O$ 0.02 mg/L, biotin 0.05 mg/L, and supplemented with 40 g/L of glucose and 0.2 mg/L of isoleucine. The cells were cultivated in triplicate at 30°C and shaken at 220 rpm for 72 h in shake flasks. The samples were harvested at 12-h intervals for assay.

Fed-batch fermentation was carried out in a 5-L bioreactor (Shanghai Bailun Bio-Technology Co., China) with a working volume of 2 L of CGX medium containing 40 g/L of glucose and 0.2 mg/L of isoleucine. Cells were cultivated in 1-L shake flask containing 200 mL of CGIII medium at 30°C for 12–16 h to reach an optimal density of approximately 12. Then, the seed cultures were inoculated into the 5-L bioreactor containing 1.8 L of CGX medium. The glucose reservoir was fed into the bioreactor at a rate of 0.2–0.5 mL/min to maintain the glucose concentration above 10 g/L in the fermentation process. The temperature was maintained at 30°C using cooling water circulation. Ammonia hydroxide was used to control the pH to 7.0–7.2 during the fermentation process. The dissolved oxygen was determined using a $pO_2$ electrode and maintained above 30% saturation level by variation of the stirrer speed.

## Genetic manipulation

All DNA manipulations were performed using standard procedures as described previously (84). Plasmids used in this study are listed in Table S9. Primers used for PCR amplification are shown in Table S9. A markerless homologous recombination system carrying the *sacB* gene was used for gene deletion/integration, and a TCCRAS system was applied for promoter replacement in the genome (85, 86). In brief, the upstream and downstream fragments or those together with promoter/gene were amplified using Q5 high-fidelity DNA polymerase and mixed with the enzyme-digested pK18*mobsacB* or pWYE587 by Gibson Assembly using a NovoRec plus One step PCR Cloning Kit (Novoprotein, China) at 50°C for 1 h and then transformed to the competent cells of EC135. The pK18*mobsacB* and pWYE587 derivatives were transformed into *C. glutamicum* by electroporation to screen the recombinant mutant via two recombination events as described previously (67, 85).

To construct the "bifido shunt", codon-optimized *fxpK* gene derived from *Bifidobacterium adolescentis* was synthesized by Sangon Biotech, Inc. (Shanghai, China). The *leuA*$^r$ and *fxpK* were ligated into pXMJ19 for the $P_{tac}$-mediated inducible expression and ligated to pZUK for the constitutive expression under the control of $P_{glyA}$ and $P_{tuf}$ by Gibson Assembly. For *alaD* overexpression, codon-optimized *alaT* gene derived from *Lysinibacillus sphaericus* was synthesized and ligated into *XbaI*/*Eco*RI-digested pXMJ19 to construct p*alaD*. For *ilvBNCDE* overexpression, *ilvBN*, *ilvC*, *ilvD,* and *ilvE* were amplified and ligated into *XbaI*/*Eco*RI-digested pXMJ19 to construct p*ilvBNCDE* by Gibson Assembly. When needed, 1 mM of isopropyl-β-D-thiogalactoside was added to the culture media to induce gene expression.

## Constraint-based metabolic flux analysis

*In silico* analysis of metabolic fluxes to overproduce targets was performed with MatLab 2014a (The MathWorks, Natick, MA, USA) and the COBRA Toolbox 2.05 (MathWorks Inc.) with *glpk* solvers, using the genome-scale metabolic model of *C. glutamicum*, *i*CW773 (45). For flux calculation, the glucose uptake flux was limited to a maximum of 4.67 mmol/gCDW/h, and the oxygen uptake flux was unrestricted. FBA, gFBA, and FVA were used to simulate fluxes for overproducing L-alanine, L-leucine, or L-valine with 20% biomass constraint, respectively. Based on the entire range of feasible minimal and maximal flux through each reaction by FVA calculation and the obtained fluxes

from FBA and gFBA, a reaction essentiality analysis was performed to identify reactions as being either essential (a non-zero flux with the same value of minimum and maximum), substitutable (the range of possible fluxes span zero), or blocked (minimum and maximum flux of zero) for target product synthesis. A simulation of the impact of aconitase flux on biomass and cellular metabolic flux distributions was performed by decreasing relative ACN flux from 1.0 to 0.1 by FBA. The FxpK-based reactions were added to the model to construct a bifido shunt pathway and simulate biomass formation and L-leucine synthesis by FBA.

## The construction and screening of a cistronic library

The first 30 nucleotides of *tsf*, *argG*, *fumC*, *mqo*, *guaA*, *pyrE*, *coaD*, *pyk*, and *rncS* genes were synthesized in primers to amplify the $P_{tuf}$-cistron (Table S9). The $P_{tuf}$-cistron and *gfp* gene were ligated to pXMJ19 to construct the series of $P_{tuf}$-cistron-GFP report plasmids. Nine report plasmids were transformed into *C. glutamicum* ATCC13032 by electroporation. The resulting strains were cultivated in BHIS at 30°C for 16 h. Then, the cells were collected and suspended in 50 mM PBS buffer (pH 7.2) for fluorescence detection. The green fluorescence intensity was evaluated by a FACS Calibur flow cytometry system (BD Biosciences, USA).

The p$P_{tuf}$-*tsf*-*gfp* plasmid was used as a template for a cistronic library construction via PCR with primers containing degenerate bases (Table S9). The PCR product was self-ligated and transformed into EC135. All the transformants were collected and used for plasmid extraction. The plasmid library was transformed into *C. glutamicum* by electroporation. A total of 2,880 monoclones harboring mutations in the M1, M2, and M3 regions of TSF were inoculated into a 96-well deep plate containing 1 mL of BHIS medium for 16 h cultivation at 30°C and 220 rpm. The cells were collected and suspended in a 96-well microplate. GFP intensity was detected by a microplate reader (Tecan, Switzerland) with excitation at 380 nm and emission between 420 and 460 nm.

Three RBS2 with different strengths were synthesized in primers and combined with 24 cistrons to construct a cistronic variant library via PCR with corresponding primers (Table S9). The PCR product was self-ligated and transformed into EC135. Seventy-two plasmids were individually extracted and transformed into *C. glutamicum* by electroporation. After cultivation in a 96-well deep plate at 30°C, cells were collected and suspended in a 96-well microplate for the second round of screening using a microplate reader (Tecan, Switzerland).

To co-overexpress *leuA*[r] and *fxpk* in a bicistronic expression cassette, $P_{glyA}$ and *leuA*[r] were amplified by overlap PCR and ligated to pZUK to construct p*leuA*[r]. The $P_{tuf}$-*tsf(D10)*-*RBS2-3*, $P_{tuf}$-*tsf(H4)*-*RBS2-3*, $P_{tuf}$-*tsf(H7)*-*RBS2-3*, $P_{tuf}$-*tsf(G6)*-*RBS2-3*, and $P_{tuf}$-*tsf(G9)*-*RBS-3* were amplified using the corresponding plasmids from cistron variant library as template by PCR and assembled with *fxpk* and p*leuA*[r] to construct p*leuA*[r]-$P_{tuf}$-*tsf(M)*-*RBS2-3*-*fxpk* derivatives.

## Gibbs free energy (ΔG) calculation

To understand the potential for forming the mRNA structure of leader cistrons, UNAfold software was used to predict the minimum-folding-energy structure conformation (87). The junction region comprises the positions between −41 and +33 in 10 leader cistrons with respect to the translation start site. The correlation analysis was performed between minimum free energy and the $\log_2$fluorescence. To evaluate the affinity between the leader cistron sequences and the 16S rRNA, the hybridization energy for each resulting RNA duplex was calculated by UNAfold using the region spanning from positions −41 to −1 nt with respect to the translation start site. The correlation analysis was performed between hybridization energy and the $\log_2$fluorescence.

## Construction of T7 RNAP-mediated sRNA

The pXMJ19 was used as a sRNA expression vector. $P_{J23119}$, $P_L$, and $P_{glyA}$ promoters were used to mediate *gfp* expression in *E. coli* and *C. glutamicum,* respectively. In the case of anti-sRNA, target-binding sequences (24 bp) were designed to complement the coding sequence that spans the AUG to nucleotide +21 of the target gene using MFold (88), and the scaffold nucleotides of MicC, DsrA, and MicF were synthesized by Sangon Biotech, Inc. (Shanghai, China). Anti-sRNA was under the control of $P_{tac}$ or $P_{T7}$ promoter. Cassettes expressing anti-sRNA and GFP were amplified by overlap PCR with the corresponding primers and ligated into *Eco*RI/*Pst*I-digested plasmids. To reduce host toxicity and maintain the activity of T7 RNAP (89), a mutation (R632S) was introduced to T7 RNAP to mediate the transcription of anti-sRNA, and T7 RNAP* was ligated into *Hind*III/*Pst*I-digested corresponding plasmids. T7 RNAP* expression was induced by the addition of IPTG. Six $P_{T7}$ variants (C4, 3F2, 3H5, 4A6, H10, and H9) were used to mediate anti-*gfp* sRNA transcription. Using RBS Calculator v2.0, the optimized RBSs with various translation initiation rates were designed to mediate T7 RNAP* translation. GFP intensity was determined by flow cytometry using a FACS Calibur flow cytometry system (BD Biosciences, USA) with excitation at 380 nm and emission between 420 and 460 nm. The anti-*aceE* and anti-*gltA* sRNAs mediated by the $P_{T7}$ variants were synthesized and ligated into the *Eco*RI/*Pst*I-digested p*T7RNAP** to construct the series of p*T7 RNAP** derivatives harboring anti-*aceE* sRNA and anti-*gltA* sRNAs (Table S9), which were transformed into *C. glutamicum* by electroporation.

## Analytical methods

Cell concentration was determined by measuring the absorbance at 600 nm ($OD_{600}$) using a spectrophotometer (V-1100D; Mapada Instruments, Shanghai, China). The cell dry weight (CDW) per liter was calculated using an experimentally determined formula: CDW (g $L^{-1}$) = 0.27 × $OD_{600}$ (90). The glucose concentration was measured using an SBA-40D biosensor analyzer (Institute of Biology of Shandong Province Academy of Sciences, Shandong, China). Amino acids in the culture supernatant were determined using high-performance liquid chromatography with a ZORBAX Eclipse AAA column (5 µm, 3.0 × 150 mm Agilent, USA) at 40℃ after online derivatization with *o*-phthalaldehyde (OPA). One microliter of the sample was mixed with 5 µL of 0.40 M borate buffer. Following the addition of 1 µL of the OPA agent, the mixture was injected into HPLC. Mobile phase A was 10 mM $Na_2HPO_4$ and $Na_2B_4O_7$ (pH 8.2), whereas mobile phase B consisted of acetonitrile, methanol, and $H_2O$ (vol:vol:vol, 45:45:10). The elution was performed using a gradient of mobile phases A and B at 1 mL/min. Elution gradients were 0–0.34 min, 2% B; 0.34–13.4 min, a linear gradient of B from 2% to 57%; 13.5–15.7 min, 100% B; and 15.8–18 min, 2% B. The eluate was monitored at 338 nm by a UV detector. Organic acids were determined by high-performance liquid chromatography equipped with an SB-Aq column (4.6 × 250 mm; Agilent Technologies, USA) at 210 nm. Mobile phase C (20 mM $KH_2PO_4$, pH 2.3) and mobile phase D (acetonitrile) were at a ratio of 95:5.

## Enzymatic activity assay

The phosphoketolase assay determines the amount of AcP formed from F6P by the ferric acetyl hydroxamate method (57). The buffer consisted of 50 mM Tris (pH 7.5), 5 mM $MgCl_2$, 5 mM potassium phosphate, and 1 mM thiamine pyrophosphate. The assay was initiated with 10 mM F6P and stopped by adding 60 µL of 2 M hydroxylamine (pH 6.5) to 40 µL of assay solution. After incubation at 30℃ for 10 min, the coloring reagent (consisting of 40 µL $FeCl_3$ in 0.1 M HCl) was added. The formed ferric acetyl-hydroxamate was measured at 505 nm. Commercial lithium potassium AcP was used to relate absorbance to concentration; the standard curve showed a linear relationship between 0 and 15 mM in the conditions described. IDH activity was assayed at 30℃ in 1 mL of a solution of 100 mM triethanolamine buffer (pH 7.6), 0.8 mM $MnSO_4$, 0.5 mM NADP,

and using 0.8 mM isocitrate as a substrate (91). The increase in NADPH was monitored at 340 nm. One unit of IDH activity is defined as 1 µmol of NADPH formed per minute. ICL activity was recorded following NADH oxidation at 340 nm. The reaction mixture contained 40 mM HEPES buffer (pH 7.0), 6 mM $MgCl_2$, 45 U lactate dehydrogenase, 0.28 mM NADH, 2 mM isocitric acid, and enzyme extract in a final volume of 1 mL (92). One unit of ICL activity is defined as 1 µmol of NAD formed per minute. MS assay was performed based on the consumption of acetyl-CoA with a decrease in the absorbance (93). One unit of MS activity is defined as 1 µmol of malate formed per minute.

## RNA preparation and quantitative real-time RT-PCR

*C. glutamicum* strains were grown to the exponential phase in the CGX minimal medium as described above. The cells were harvested, and the total RNA was isolated using the RNAprep Pure Cell/Bacteria Kit (Tiangen, China). Reverse transcription of approximately 400 ng of RNA was performed with the specific primers listed in Table S9 using the FastQuant RT Kit (Tiangen, China). Quantitative PCR was performed with the GoTaq qPCR master mix (Promega, USA) in a 20 µL mixture using the LightCycler 96 Real-Time PCR System (Roche, Switzerland). The *rpoB* gene was used as the reference gene to normalize the mRNA levels of *gltA*, *acn*, *icd*, *odhA*, *aceA*, *aceB*, *zwf*, *tal*, *tkt,* and *pta* genes. The qPCR products were verified via a melting curve analysis. Data collection and analyses were conducted using the LightCycler 96 software (Roche, Switzerland) according to the $2^{-\Delta\Delta CT}$ method (94).

## Adaptive laboratory evolution

ALE was conducted with the LUE-28 strain in 250-mL shaking flasks. The seed cultivation was performed in a 250-mL conical flask with a 25 mL BHI medium overnight at 30°C. For ALE cultivations, the modified CGX minimal medium containing 20 g/L glucose as a carbon source and 0.2 mg/L L-isoleucine was supplemented with 1 g/L yeast extract. The concentration of yeast extract was decreased gradually in the order of 1, 0.8, 0.6, 0.4, 0.2, and 0 g/L to evolve a strain grown on glucose as a sole carbon source. The evolution experiment started with six biological replicates of each strain and an initial cell density ($OD_{600}$) of 2. Cells were cultivated for 24 h at 30°C on a rotary shaker at 220 rpm, and then an aliquot of the culture was transferred to a new flask for another round of growth. This serial passage in flasks allowed a time interval of about 7 days for the evolved cells with more fitness to lower concentration of yeast extract and till 0 g/L of yeast extract. At defined time points (approximately every sixth transfer), samples were taken and stored as glycerol cultures.

## Optical analysis of cells and cell size determination

*C. glutamicum* strains were grown at 30°C in CGX medium and harvested in the exponential phase. After washing with PBS buffer twice, the cell suspension was diluted to an $OD_{600}$ of 0.2 on glass slides. Microscopic investigation and cell size determination were performed with a Zeiss Axio Imager A1 Microscope equipped with a transmitted light illuminator and a high-resolution Zeiss Axio Cam camera. Imaging was acquired with Zeiss Axio Vision imaging software (Carl Zeiss MicroImaging GmbH, Jena, Germany). Cell sizes were determined with a computationally efficient image analysis tool MicrobeJ on the platform Image J (95).

## Genome sequencing analysis

Genomic DNAs from five evolved strains and the initial LEU-28 strain were extracted with the SDS method. Sequencing libraries were generated using NEBNext Ultra DNA Library Prep Kit for Illumina (NEB, USA). The whole genome was sequenced using Illumina NovaSeq PE150. The original figure data were transformed into raw sequenced reads by CASAVA base calling. The sequenced data were filtered, and the sequence of adapter and low-quality data were removed. Sequence alignment was performed by

mapping the reads to the reference sequence of *C. glutamicum* genome (NCBI accession: NC_003450.3) using BWA software (V0.7.8) to identify nucleotide mutations and annotate gene function.

## Transcriptome analysis

RNA-seq data were generated from cell cultures under the exponential growth phase. Cell pellets were stored at −80°C before RNA extraction. Total RNA was isolated and purified using a Total RNA Extractor (Trizol) extraction kit (Sangon, Shanghai, China). Total RNA quality was checked using Qubit2.0 and agarose gel electrophoresis. rRNA was removed using the Ribo-off rRNA Depletion Kit (Vazyme, Nanjing, China). RNA-seq libraries were generated using VAHTS Stranded mRNA-seq Library Prep Kit for Illumina. RNA-seq libraries were run on an Illumina NovaSeq6000. Raw sequencing reads in FastQ format were first mapped to the reference genome (NCBI accession: NC_003450.3) using Bowtie2 and evaluated by RSeQC in Python. The abundance of the transcript was obtained using the featureCounts and standardized by transcripts per kilobase million (TPM).

## Targeted metabolomic analysis

An ultra-performance liquid chromatography coupled to tandem mass spectrometry (UPLC-MS/MS) system (ACQUITY UPLC-Xevo TQ-S, Waters Corp., Milford, MA, USA) was used to quantitate targeted metabolites in Novogene Co., Ltd. (Beijing, China). Samples were injected into an ACQUITY UPLC BEH C18 1.7 µm VanGuard pre-column (2.1 × 5 mm) and an ACQUITY UPLC BEH C18 1.7 µm analytical column (2.1 × 100 mm) using a 18-min linear gradient at a flow rate of 0.4 mL/min for the positive/negative polarity mode. The eluents were eluent A (0.1% formic acid-water) and eluent B (acetonitrile: IPA = 70:30). The solvent gradient was set as follows: 0–1 min (5% B), 1–11 min (5%–78% B), 11–13.5 min (78%–95% B), 13.5–14 min (95%–100% B), 14–16 min (100% B), 16–16.1 min (100%–5% B), and 16.1–18 min (5% B). The Xevo TQ-S mass spectrometer was operated in positive (negative) polarity mode with a capillary of 1.5 (2.0) kV, a source temperature of 150 °C, a desolvation temperature of 550°C, and a desolvation gas flow of 1,000 L/h. The detection of the experimental samples using multiple reaction monitoring was based on Novogene self-built method. The Q1, Q3, retention time, declustering potential, and collision energy were used for metabolite identification. The ratio of the Q3 peak area of the compound to the internal standard was brought into the standard curve. The concentration of the compound was calculated from the known internal standard concentration. The data files generated by UPLC-MS/MS were processed using the MassLynx Version 4.1 to integrate and correct the peak.

## Docking analysis

Molecular docking was performed with the AutoDock tools v1.5.4 and AutoDock v4.2 program from the Scripps Research Institute (http://www.scripps.edu/mb/olson/doc/autodock). Isocitrate was docked to the model of IDH (PDB No. 2b0t). Receptor-ligated interaction was shown using Pymol 2.5.2 software.

## Statistics

Data from three independent experiments are presented as the mean ± S.D. The two-tailed unpaired Student's *t*-test was conducted to determine the significance using GraphPad Prism 7.0 (San Diego, CA, USA). $P < 0.05$ was considered statistically significant.

## ACKNOWLEDGMENTS

We are grateful to Tong Zhao for the flow cytometry analysis and Weilin Li for metabolite detection.

This work was supported by grants from the National Natural Science Foundation of China (32071460, 31870074) and the National Key R&D Program of China (2021YFC2101800).

Y.Z. and T.W. conceptualized the study. Y.Z., X.W., C.O., Q.H., and D.L. designed the methodology. Y.Z., X.W., C.O., Q.H., D.L., Y.T., and Z.L. were involved in the investigation. Y.Z. and X.W. performed data analysis. J.M. and S.L. assisted with the study. T.W. supervised the study. Y.Z. wrote the manuscript. Y.Z. and T.W. reviewed and edited the manuscript. All authors commented on the final manuscript.

## AUTHOR AFFILIATIONS

[1]State Key Laboratory of Microbial Resources, Institute of Microbiology, Chinese Academy of Sciences, Beijing, China
[2]College of Life Sciences, University of Chinese Academy of Sciences, Beijing, China
[3]Savaid Medical School, University of Chinese Academy of Sciences, Beijing, China

## AUTHOR ORCIDs

Yun Zhang http://orcid.org/0000-0002-2617-8080
Xueliang Wang http://orcid.org/0000-0003-3332-9032
Tingyi Wen http://orcid.org/0000-0003-2583-7088

## FUNDING

| Funder | Grant(s) | Author(s) |
| --- | --- | --- |
| MOST \| National Natural Science Foundation of China (NSFC) | 32071460, 31870074 | Yun Zhang |
| MOST \| National Key Research and Development Program of China (NKPs) | 2021YFC2101800 | Tingyi Wen |

## AUTHOR CONTRIBUTIONS

Yun Zhang, Conceptualization, Data curation, Funding acquisition, Investigation, Visualization, Writing – original draft, Writing – review and editing | Xueliang Wang, Investigation, Methodology, Software | Christianah Odesanmi, Investigation | Qitiao Hu, Investigation | Dandan Li, Investigation, Methodology | Yuan Tang, Investigation | Zhe Liu, Investigation | Jie Mi, Investigation | Shuwen Liu, project administration | Tingyi Wen, Conceptualization, Funding acquisition, Supervision, Writing – review and editing

## DATA AVAILABILITY

Transcriptomic raw data of ALE0 and LEU-28 strains can be accessed from the NCBI Short Read Archive with BioProject ID SRR23280551 and SRR23280552, respectively. The mutations in the genome of ALE strains are available in Table S6. Metabolomic raw data are available in Table S8. The data supporting the findings of this work are available from the corresponding author upon reasonable request.

## ADDITIONAL FILES

The following material is available online.

### Supplemental Material

**Supplemental Material (mSystems00839-23-s0001.docx).** Fig. S1-S12.
**Table S1 (mSystems00839-23-s0002.xlsx).** Flux distribution for maximum of biomass, alanine, leucine, and valine by FBA.
**Table S2 (mSystems00839-23-s0003.xlsx).** Constraint-based FBA, gFBA, and FVA analyses.

**Table S3 (mSystems00839-23-s0004.xlsx).** Flux distribution in response to *pyc* deletion.

**Table S4 (mSystems00839-23-s0005.xlsx).** Fermentation parameters of LEU strains.

**Table S5 (mSystems00839-23-s0006.xlsx).** Flux distribution in response to the introduction of the bifido shunt pathway.

**Table S6 (mSystems00839-23-s0007.xlsx).** Genome mutations in the different evolved strains.

**Table S7 (mSystems00839-23-s0008.xlsx).** Comparative transcriptomic analysis between LEU-28 and ALE0 strains.

**Table S8 (mSystems00839-23-s0009.xlsx).** Comparative metabolic analysis between LEU-28 and ALE0 strains.

**Table S9 (mSystems00839-23-s0010.xlsx).** Strains, plasmids, and primers.

## Open Peer Review

**PEER REVIEW HISTORY (review-history.pdf).** An accounting of the reviewer comments and feedback.

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
