## [Reviewer comments · mSystems]

Model-guided metabolic rewiring to bypass pyruvate oxidation for pyruvate derivative synthesis by minimising carbon loss

Yun Zhang, Xueliang Wang, Christianah Odesanmi, Qitiao Hu, Dandan Li, Yuan Tang, Zhe Liu, Jie Mi, Shuwen Liu, and Tingyi Wen

Corresponding Author(s): Tingyi Wen, Institute of Microbiology, Chinese Academy of Sciences

Review Timeline:

Submission Date:	August 9, 2023
Editorial Decision:	December 15, 2023
Revision Received:	January 4, 2024
Accepted:	January 8, 2024

Editor: Jack Gilbert

Reviewer(s): The reviewers have opted to remain anonymous.

Transaction Report:

DOI: <https://doi.org/10.1128/msystems.00839-23>

Re: mSystems00839-23 (Model-guided metabolic rewiring to bypass pyruvate oxidation for pyruvate derivative synthesis by minimising carbon loss)

Dear Dr. Tingyi Wen:

Thank you for the privilege of reviewing your work. Below you will find instructions from the mSystems editorial office, and the reviewer comments.

Revision Guidelines

Sincerely,
Jack Gilbert
Editor
mSystems

Reviewer #3 (Comments for the Author):

The authors in this work aimed to decouple pyruvate metabolism from cell growth to improve pyruvate availability which can be used to derive products of interest. Metabolic modelling and laboratory evolution guided rational metabolic engineering led to develop a chassis *C. glutamicum* strain by introducing a non-pyruvate-derived acetyl-CoA synthetic pathway [bifido shunt] to bypass pyruvate oxidation. Also the strain was rewired for C4 metabolism by activating the glyoxylate shunt and attenuating TCA cycle under aerobic condition for glucose catabolism by introducing a mutated IDH. They adopted and demonstrated genome-scale metabolic model of *C. glutamicum* combined with FBA, gFBA and FVA analyses. The specific growth rate and

biomass increased in response to the introduction of bifido shunt pathway. Growth-related regulatory targets were screened using ALE and multi-omics analysis. Further gene expressions were fine-tuned with genetic control tools and demonstrated the downregulation of the TCA cycle by a T7 RNAP-mediated sRNA switch. Finally, the authors demonstrated that the engineered bacteria with the glyoxylate shunt as only C4 anaplerotic pathway and the bifido shunt as the non-pyruvate-derived acetyl-CoA synthetic pathway maintains growth fitness and minimises carbon loss under aerobic condition in glucose fed cells there by enhancing the flux from pyruvate to enhance the targeted yield of L-leucine, L-alanine and L-valine in fed batch fermentation in bioreactor.

The manuscript needs improvement with following revisions.

1. Abstract- should explicitly reflect what are the three algorithm analysis? Constraint based FBA, gFBA and FVA analyses?
2. "a new non-pyruvate-derived acetyl-CoA synthetic pathway is introduced to bypass pyruvate oxidation", - can introduce bifido shunt in abstract as a promising route with its influence on carbon utility quantitatively
3. The advantage of having Glyoxylate shunt as only C4 anaplerotic pathway can be described is the abstract better
4. Would advise rewriting the abstract which can also reflect the yields achieved for L-leucine and other aminoacids.
5. Can the authors compare and comment on Carbon Usage Efficiency among all the engineered strains in this study [when Glucose fed and aerobic ofcourse]
6. Why fermentation profile of the LEU-29 and engineered chasis look same? Check Figure 7D and 7F and make necessary revisions.
7. Font sizes of the text in the figures are not easily legible and hence can be increased where possible. Font size in Figure# 1D-1H, 2A-2D, 3B-3E, 4, 6 is not clear/legible.
8. Please revise promoters mentions in text. One thorough reading of text will help.
9. Results and Discussion are well written but results of library prep and screening can be discussed in detail. As figure text is not clear cant co-relate with results.
10. Few lines in discussions can be rephrased or split into two e.g., Line# 561- 567, 574, 581.
11. Citation can be added in line # 649.
12. Few lines need revision in punctuations or Grammer: line# 91, 126, 331, 488, 521, 557, 572.
13. Unit's representation in materials and methods like hours or hrs or h needs consistency
14. Heading in materials and methods should be consistent line# 687.
15. Citation can be added in line # 649.
16. Ensure all the abbreviated enzymes have full forms in text
17. Revision in few words can be considered e.g., line # 43: different - difficult? One thorough reading of text will help.
18. line #684 fluxed?

Model-guided metabolic rewiring to bypass pyruvate oxidation for pyruvate derivative synthesis by minimising carbon loss

mSystems00839-23

Reviewer comments:

The authors in this work aimed to decouple pyruvate metabolism from cell growth to improve pyruvate availability which can be used to derive products of interest. Metabolic modelling and laboratory evolution guided rational metabolic engineering led to develop a chassis *C.glutamicum* strain by introducing a non-pyruvate-derived acetyl-CoA synthetic pathway [bifido shunt] to bypass pyruvate oxidation. Also the strain was rewired for C4 metabolism by activating the glyoxylate shunt and attenuating TCA cycle under aerobic condition for glucose catabolism by introducing a mutated IDH. They adopted and demonstrated genome-scale metabolic model of *C.glutamicum* combined with FBA, gFBA and FVA analyses. The specific growth rate and biomass increased in response to the introduction of bifido shunt pathway. Growth-related regulatory targets were screened using ALE and multi-omics analysis. Further gene expressions were fine-tuned with genetic control tools and demonstrated the downregulation of the TCA cycle by a T7 RNAP-mediated sRNA switch. Finally, the authors demonstrated that the engineered bacteria with the glyoxylate shunt as only C4 anaplerotic pathway and the bifido shunt as the non-pyruvate-derived acetyl-CoA synthetic pathway maintains growth fitness and minimises carbon loss under aerobic condition in glucose fed cells there by enhancing the flux from pyruvate to enhance the targeted yield of L-leucine, L-alanine and L-valine in fed batch fermentation in bioreactor.

The manuscript needs improvement with following revisions.

1. Abstract- should explicitly reflect what are the three algorithm analysis? Constraint based FBA, gFBA and FVA analyses?
2. “a new non-pyruvate-derived acetyl-CoA synthetic pathway is introduced to bypass pyruvate oxidation”, - can introduce bifido shunt in abstract as a promising route with its influence on carbon utility quantitatively
3. The advantage of having Glyoxylate shunt as only C4 anaplerotic pathway can be described is the abstract better
4. Would advise rewriting the abstract which can also reflect the yields achieved for L-leucine and other aminoacids.

5. Can the authors compare and comment on Carbon Usage Efficiency among all the engineered strains in this study [when Glucose fed and aerobic ofcourse]
6. Why fermentation profile of the LEU-29 and engineered chassis look same? Check Figure 7D and 7F and make necessary revisions.
7. Font sizes of the text in the figures are not easily legible and hence can be increased where possible. Font size in Figure# 1D-1H, 2A-2D, 3B-3E, 4, 6 is not clear/legible.
8. Please revise promoters mentions in text. One thorough reading of text will help.
9. Results and Discussion are well written but results of library prep and screening can be discussed in detail. As figure text is not clear cant co-relate with results.
10. Few lines in discussions can be rephrased or split into two e.g., Line# 561- 567, 574, 581.
11. Citation can be added in line # 649.
12. Few lines need revision in punctuations or Grammer: line# 91, 126, 331, 488, 521, 557, 572.
13. Unit's representation in materials and methods like hours or hrs or h needs consistency
14. Heading in materials and methods should be consistent line# 687.
15. Citation can be added in line # 649.
16. Ensure all the abbreviated enzymes have full forms in text
17. Revision in few words can be considered e.g., line # 43: different – difficult? One thorough reading of text will help.
18. line #684 fluxed?

Response to Reviewers

Dear Prof. Gilbert,

Enclosed please find the revised version of our manuscript entitled “Model-guided metabolic rewiring to bypass pyruvate oxidation for pyruvate derivative synthesis by minimising carbon loss” (mSystems00839-23). We appreciated very much for those valuable comments and helpful suggestions from the reviewers, which have guided us to significantly improve the quality of our manuscript. According to the reviewer's comments, we have thoroughly revised the manuscript. In addition, the point-by-point responses to the comments are listed below.

Responses to Reviewer #3's comments

1. Abstract- should explicitly reflect what are the three algorithm analysis? Constraint based FBA, gFBA and FVA analyses?

Response: As suggested, the description of three algorithm analysis has been shown in abstract as following: “a genome-scale metabolic model combined with constraint-based flux balance analysis, geometric flux balance analysis, and flux variable analysis were used to identify genetic targets for strain design.”

2. "a new non-pyruvate-derived acetyl-CoA synthetic pathway is introduced to bypass pyruvate oxidation", - can introduce bifido shunt in abstract as a promising route with its influence on carbon utility quantitatively.

Response: As suggested, a bifido shunt has been supplemented in abstract as following: “a bifido shunt pathway was introduced to generate 3 molecules of non-pyruvate-derived acetyl-CoA from 1 molecule of glucose, bypassing pyruvate oxidation and carbon dioxide generation.”

3. The advantage of having Glyoxylate shunt as only C4 anaplerotic pathway can be described is the abstract better.

Response: The advantage of having glyoxylate shunt as only C4 anaplerotic pathway

has been described in abstract as following: “a mutated isocitrate dehydrogenase functioned as a metabolic switch to activate the glyoxylate shunt as only C4 anaplerotic pathway to generate malate from two molecules of acetyl-CoA input and bypass two decarboxylation reactions in the tricarboxylic acid cycle.”

4. Would advise rewriting the abstract which can also reflect the yields achieved for L-leucine and other amino acids.

Response: As suggested, the titers of L-leucine, L-alanine and L-valine have been supplemented in abstract as following: “A chassis strain for pyruvate derivative synthesis was constructed to reduce carbon loss by using the glyoxylate shunt as only C4 anaplerotic pathway and the bifido shunt as a non-pyruvate-derived acetyl-CoA synthetic pathway, and produced 22.46, 27.62 and 6.28 g/L of L-leucine, L-alanine and L-valine by a controlled sRNA switch, respectively.”

5. Can the authors compare and comment on Carbon Usage Efficiency among all the engineered strains in this study [when Glucose fed and aerobic of course]

Response: Thank for your suggestion. According to the mole conversion coefficient (0.095) by experimental detection, we recalculate the yield of biomass from glucose, which represents that mole number of biomass generating from per mole of glucose. The total yield of biomass and L-leucine from glucose is used to represent carbon usage efficiency and all values of total yield for the engineered strains are added to Supplementary Table S4. The carbon usage efficiencies of engineered strains are compared in the case of the bifido shunt introduction and ALE in the revised manuscript as following:

“Remarkably, the total yield of biomass and L-leucine from glucose increased by 17% and 16% in LEU-12 and LEU-13 strains respectively (Table S4), indicating that the introduction of bifido shunt pathway significantly increases carbon usage efficiency by decreasing carbon dioxide generation.”

“The total yield of biomass and L-leucine from glucose in ALE0 strain increased by 20% compared to that in LEU-27 strain (Table S4), indicating that metabolic

reprogramming during ALE improves carbon usage efficiency.”

6. Why fermentation profile of the LEU-29 and engineered chassis look same? Check Figure 7D and 7F and make necessary revisions.

Response: It is a mistake for us to use the same graph to show the fermentation profile of the LEU-29 strain. Figure 7D is corrected with the right graph of the fermentation profile in the revised manuscript.

7. Font sizes of the text in the figures are not easily legible and hence can be increased where possible. Font size in Figure# 1D-1H, 2A-2D, 3B-3E, 4, 6 is not clear/legible.

Response: As suggested, font sizes of the text in the figures are magnified to be legible for readers. Figures 1, 2, 3, 4 and 6 are modified to be more clearly.

8. Please revise promoters mentions in text. One thorough reading of text will help.

Response: All promoters in text have been shown in a unified format throughout the revised manuscript.

9. Results and Discussion are well written but results of library prep and screening can be discussed in detail. As figure text is not clear cant co-relate with results.

Response: As suggested, the results of library preparation and screening have been further analyzed in discussion as following: “When comparing the impacts of 10 leader cistrons on fluorescence intensity, we observe that translation efficiency mainly depends on the folding energy of the leader cistron, which is consistent with the observation in *E. coli* (55). Unexpectedly, BCD2 as a leader cistron produced a lower level of fluorescence intensity in *C. glutamicum* than that in *E. coli*, which might attribute to the codon usage bias in the different species (56). To amplify the regulatory range of leader cistron on translation efficiency, 9 nucleotide variance in three regions of Tsf leader cistron (M1, M2 and M3) can theoretically generate about 4^9 mutants, which is lower than that arose from 18 nucleotide variance. It can not only cover a desired expression parameter space but also reduce library size and screening

workload. Finally, we obtained mutant cistronic library covering a 76-fold range.”

The figure legends have been rewritten to be correlated with the descriptions in results.

10. Few lines in discussions can be rephrased or split into two e.g., Line# 561- 567, 574, 581.

Response: As suggested, the mentioned sentences have been rephrased and divided into two paragraphs as following: “Ser113 residue is hidden deep within a pocket and points away from the surface of Apo-ICD (72). However, the interaction between AceK and ICD triggers a conformational change of ICD, resulting in Ser113 adopting an exposed, outward-facing position to be more accessible for phosphotransfer (73).

Dimeric ICD is regulated through the phosphorylation cycle catalyzed by AceK. Different from ICD, the comparison of substrate- and NADP⁺-binding IDH structures demonstrates that the binding of isocitrate or NADP⁺ to IDH induces a domain shift, resulting in a more closed conformation at the active site (74).”

11. Citation can be added in line # 649.

Response: The reference has been cited in Line 655.

12. Few lines need revision in punctuations or Grammer: line# 91, 126, 331, 488, 521, 557, 572.

Response: The mentioned sentences have been revised as following:

“and generates a lipoic acid or acetate auxotroph strain.” (Line 91)

“ALE can rapidly generate desired phenotypes by enriching mutations on metabolic enzymes, rewiring serendipitous pathways and changing transcriptional profiles.” (Lines 124-125)

“The recombinant strains harboring two sRNAs showed the different increases in l-leucine titer and yield” (Line 327)

“The NOG pathway generating 3 molecules of acetyl-CoA from 1 molecule of glucose maximized the carbon yield without carbon dioxide loss.” (Lines 496-497)

“We developed a T7 RNAP-mediated sRNA dynamic control system, which was independent on the host’s machinery and able to switch off essential pathways in a reciprocal fashion” (Lines 528-530)

“the dimeric isocitrate dehydrogenase (ICD) was post-translationally modified through allosteric regulation in *E. coli*”(Lines 563-565)

“It supports our findings that the decreased IDH activity causes an activation of the glyoxylate shunt in *C. glutamicum* to metabolize isocitrate from glucose for cell growth.”(Lines 580-582)

13. Unit's representation in materials and methods like hours or hrs or h needs consistency

Response: All unit’s representation are consistent in materials and methods.

14. Heading in materials and methods should be consistent line# 687.

Response: Heading in materials and methods is consistent according to the standards of mSystems.

15. Citation can be added in line # 649.

Response: The reference has been cited in Line 655.

16. Ensure all the abbreviated enzymes have full forms in text.

Response: All the full forms of abbreviated enzymes have been checked in text.

17. Revision in few words can be considered e.g., line # 43: different - difficult? One thorough reading of text will help.

Response: The "different" has been corrected to "difficult". The manuscript has been revised carefully.

18. line #684 fluxed?

Response: The "fluxed" has been corrected to "flux".

Responses to Reviewer #2's comments

Major comments:

1. The authors mention they utilize three algorithm analyses (flux balance analysis (FBA), geometricFBA (gFBA), and flux variable analysis (FVA)) for strain design. These three methods although very powerful tools in constraint-based metabolic modeling, are not the state of the art computational methods for strain design (ref – Biz et al. <https://doi.org/10.1016/j.biotechadv.2019.04.001>). Please mention why these tools were preferred over the more sophisticated and popular strain design tools such as Optknock, cMCS or other computational methods that are the best computational tools used for computational strain design.

Response: Constraint-based FBA, gFBA and FVA are generally used to predict phenotypic traits, such as growth rate and reaction fluxes under various conditions. FBA shows that the fluxes for target amino acid synthesis are maximized in the premise of close-to-zero biomass yield, indicating target amino acid synthesis for strong growth coupling. Therefore, based on 20% lowest biomass constraint, FBA, gFBA and FVA are performed to identify reaction essentiality and corresponding genetic disturbance for overproducing target product. Only the reactions with minimum and maximum flux of zero need to be blocked for target product synthesis. Certain essential reactions coupling with growth can be increased or decreased based on the effect of flux variation on the synthesis of target product.

As mentioned by reviewer 2, Optknock and cMCS are sophisticated algorithms for strain design (Biz et al., 2019). Optknock can provide a set of reaction knockouts for maximizing production on a growth-based constraint. cMCS can give a list of minimal knockout sets using a minimum biomass yield as a constraint. The results of two algorithms only depict the genes that can be deleted. However, the growth-coupled genes are often essential for biomass and cannot be presented in a set of reaction knockouts by using a minimum biomass yield as a constraint. It leads to be unable to determine the genetic optimization for growth-coupled essential genes. Furthermore, the calculations of Optknock and cMCS are not as simple and fast as those of FBA, gFBA and FVA due to the computational nature of mixed-integer

linear programming problems. It also needs time-consuming iterative calculation to solve the knockout combination or minimal cut sets.

Therefore, we use three simple algorithms to not only screen knockout genes but also to determine the genetic optimization approach of essential genes for growth. Compared to Optknock and cMCS, three algorithm analyses are more comprehensive to design genetic optimization approach.

2. The authors mention they used the *C. glutamicum* genome scale metabolic model (GSMM) *iCW773* for their analysis which contains about 1207 reactions and 773 genes. The supporting data provided with this manuscript only shows the analysis for merely 10% of these reactions that belong to central metabolic pathways. Why not use a reduced model instead of a GSMM? What is the justification and assumptions, if any, for eliminating rest 90% reactions from the analysis? Also, GSMM are not artificial intelligence tools (Line 71 – 72).

Response: The *iCW773* model instead of a reduced model is used to simulate reaction fluxes by FBA, gFBA and FVA in our study. The *iCW773* model contains comprehensive reactions through GPR interactions, especially certain reactions associated with growth, which can be used to simulate the whole-cell flux distribution and to screen genetic targets for strain design. Whether a reduced model or a simplified model with eliminating rest 90% reactions cannot be used to predict reaction fluxes, because a reduced model is not enough to meet the requirements for biomass synthesis and strain design.

In the supporting data, many reactions that belong to central carbon, amino acid, and nucleotide metabolic pathways have been supplemented to Tables S1, S2, S3 and S5. The cover range of reaction increases to 30%. The remaining reactions with a zero metabolic flux are meaningless and not shown in supporting data.

The “artificial intelligence tools” is corrected to “an important technological tool”.

3. Line 184 - L-leucine titer increased by 9%, 19% and 8%, respectively (Fig. 1D and Table S4). It is not 19% but should be 10%.

Response: We carefully check the results, and confirm that L-leucine titer increased by 9%, 19% and 8%, respectively. In fact, L-leucine yields reach 0.09, 0.10 and 0.09 mol/mol glucose in the corresponding strains, respectively.

4. Line 342 – It is not clear why yeast extract which is an undefined media component used in the ALE studied. The selection pressure during ALE should be for improving the production phenotype and not just growth and self-mutation.

Response: The majority of intracellular pyruvate under aerobic growth is depleted to generate acetyl-CoA fueling TCA cycle and supporting cell growth. To decouple pyruvate catabolism from cell growth and improve pyruvate availability for leucine synthesis, two C4 anaplerotic pathways were blocked and led to abolish the aerobic growth of LEU-28 strain, generating guided evolutionary pressure using cell growth as a screening marker instead of production index. The aim is to create a metabolic scenario to achieve metabolic homeostasis between cell growth and product synthesis. The supplement of yeast extract is to support the initial growth, ensuring the sufficient availability of cells by subculture for continuous evolution to rescue growth. In the absence of yeast extract, the final ALE0 strain could grow on glucose as a sole carbon source due to self-mutation. Omic-analysis and reverse engineering revealed that the activation of glyoxylate shunt by mutated IDH rescued growth and improved carbon usage efficiency.

5. Lines 379 – 382 Authors mention that differential gene expression profiles of ALE0 and LEU-28 strains were related to carbon metabolism, amino acid biosynthesis, purine metabolism, and oxidative phosphorylation (Fig. 6E and Fig. S10A). Carbon metabolism and amino acid biosynthesis are superpathways and too generic metabolic pathways to be mentioned here. Please reword.

Response: As suggested, the sentence has been rewritten as following: “The gene expression profiles of ALE0 and LEU-28 strains were distinguished in metabolic pathways related to glycolysis, TCA cycle, pentose phosphate pathway (PPP), phenylalanine, valine, glycine, arginine and histidine metabolism as well as purine

metabolism.”

6. Authors mention during fed batch cultivation a final titer of 17.92 g/L L-leucine was achieved with a yield of 0.12 mol/mol glucose and a productivity of 0.36 g/L/h. The fed batch media contained 40 g/L of glucose and 0.2 mg/L of isoleucine. How was the yield calculated for only glucose as the substrate? Also, it is important to mention how these (titer, rate, yield) TRY values compare to the latest advancement in *C. glutamicum* Leucine producing engineered strains (ref - Vogt et al. Metab Eng. 2014;22:40–52, Huang et al. Afr J Biotechnol. 2017;16:1048–1060 and Feng et al. Molecules. 2018;23:2102).

Response: In the strategy for engineering L-leucine, L-alanine, and L-valine producers, *ilvA* was deleted to decrease the split-flow at *ilvBC*-catalyzed reaction, resulting in isoleucine auxotroph. The supplement of trace isoleucine aims to restore cell growth. Glucose in the fed batch media is used as substrate to produce L-leucine and trace isoleucine has little effect on yield of L-leucine from glucose. The molar yield was calculated using the mole number of L-leucine divided by the mole number of total consumed glucose in the fed batch fermentation. The total consumed glucose consists of the initial glucose in the media and the fed glucose during fed batch cultivation. Finally, the LEU-29 strain produced 17.92 g/L leucine with a yield of 0.12 mol/mol glucose.

As suggested, the L-leucine titer in our study is compared with the other de novo engineered leucine-producing *C. glutamicum* strains in the results as following: “The L -leucine titer in this study is higher than that in the engineered strain overexpressing aminotransferase, and is comparable to that in the engineered strain being deficient in being deficient in two regulators *ltbR* and *iolR* (48, 49)”.

One strain mentioned by reviewer 2 is derived from an L-leucine producer created by random mutagenesis (Afr J Biotechnol. 2017,16:1048–1060), which is not comparable to the de novo-engineered strain due to unclear genetic background.

Minor comments:

1. Table S3 not cited in the main text.

Response: The Table S3 is cited in the main text.

2. Line 789 – full form for MD is missing.

Response: MD is an incorrect abbreviation, and MS is correct and has been shown in the revised manuscript.

3. Data availability – information for the genomics and metabolomics data availability should also be added in this section.

Response: As suggested, information for the genomics and metabolomics data is added to Data availability as following: “The mutations in the genome of ALE strains can be available in Supplemental Table S6. Metabolomic raw data is available in Supplemental Table S8.”

Re: mSystems00839-23R1 (Model-guided metabolic rewiring to bypass pyruvate oxidation for pyruvate derivative synthesis by minimising carbon loss)

Dear Dr. Tingyi Wen:

Your manuscript has been accepted, and I am forwarding it to the ASM production staff for publication. Your paper will first be checked to make sure all elements meet the technical requirements. ASM staff will contact you if anything needs to be revised before copyediting and production can begin. Otherwise, you will be notified when your proofs are ready to be viewed.

Featured Image Submissions: If you would like to submit a potential Featured Image, please email a file and a short legend to mSystems@asmusa.org. Please note that we can only consider images that (i) the authors created or own and (ii) have not been previously published. By submitting, you agree that the image can be used under the same terms as the published article. File requirements: square dimensions (4" x 4"), 300 dpi resolution, RGB colorspace, TIF file format.

Sincerely,
Jack Gilbert
Editor
mSystems